# FURINA: A Fully Customizable Role-Playing Benchmark via Scalable Multi-Agent Collaboration Pipeline

## Abstract

As large language models (LLMs) advance in role-playing (RP) tasks, existing benchmarks quickly become obsolete due to their narrow scope, outdated interaction paradigms, and limited adaptability across diverse application scenarios. To address this gap, we introduce FURINA-Builder, a novel multi-agent collaboration pipeline that automatically constructs fully customizable RP benchmarks at any scale. It enables evaluation of arbitrary characters across diverse scenarios and prompt formats, as the first benchmark builder in RP area for adaptable assessment. FURINA-Builder simulates dialogues between a test character and other characters drawn from a well-constructed character-scene pool, while an LLM judge selects fine-grained evaluation dimensions and adjusts the test character's responses into final test utterances. Using this pipeline, we build FURINA-Bench, a new comprehensive role-playing benchmark featuring both established and synthesized test characters, each assessed with dimension-specific evaluation criteria. Human evaluation and preliminary separability analysis justify our pipeline and benchmark design. We conduct extensive evaluations of cutting-edge LLMs and find that o3 and DeepSeek-R1 achieve the best performance on English and Chinese RP tasks, respectively. Across all models, established characters consistently outperform synthesized ones, with reasoning capabilities further amplifying this disparity. Interestingly, we observe that model scale does not monotonically reduce hallucinations. More critically, for reasoning LLMs, we uncover a novel trade-off: reasoning improves RP performance but simultaneously increases RP hallucinations. This trade-off extends to a broader Pareto frontier between RP performance and reliability for all LLMs. Together, these findings demonstrate the effectiveness of FURINA-Builder and the challenge posed by FURINA-Bench, establishing a strong foundation for future research on RP evaluation.

## 1 Introduction

The rapid evolution of large language models (LLMs) has spurred unprecedented advances in conversational AI agents (Shanahan et al., 2023; Chen et al., 2024; Wu et al., 2025c). Among them, role-playing conversational agents (RPCAs), widely deployed on platforms such as Character.ai[1] and Xingchen[2], have gained great attention by enabling users to engage in rich, multi-faceted dialogues with diverse characters. This breadth of applications underscores the growing need for systematic evaluation frameworks. In response, a number of role-playing (RP) benchmarks have been proposed (Tu et al., 2024; Wu et al., 2025a) to assess the RP abilities of LLMs.

However, most existing benchmarks are static and struggle to keep pace with the dynamic nature of real-world RP scenarios. These scenarios are characterized by diverse user-specified character personas, varying dialogue structures, and evolving evaluation dimensions, all of which constrain the generalizability of fixed benchmarks and make comprehensive evaluation particularly challenging. For example, existing fixed-character benchmarks (Xu et al., 2024; Wang et al., 2025b) may not align closely with target NPC designs, limiting their ability to accurately assess performance

---

[1] https://character.ai/
[2] https://tongyi.aliyun.com/xingchen/

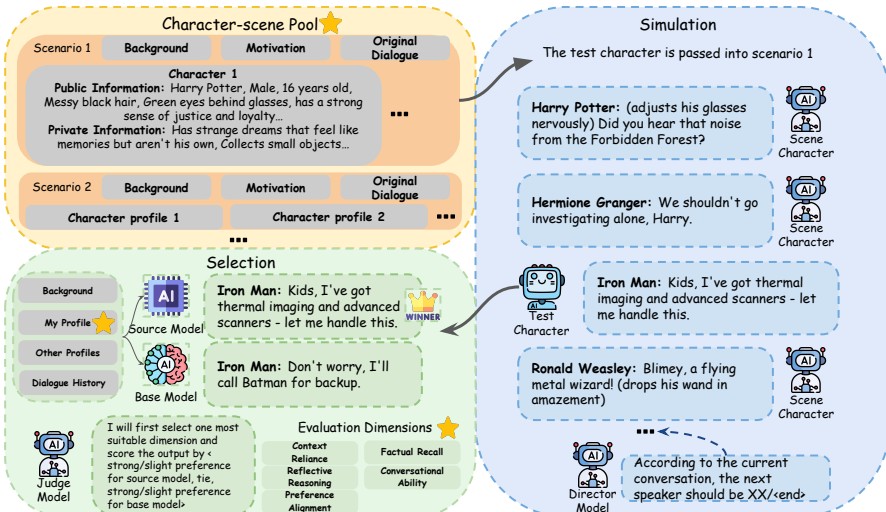

Figure 1: Overview of FURINA-Builder. There are three components. (i) Character-scene pool: a data pool containing a large number of authentic dialogue scenarios. (ii) Simulation: the test character is passed into the scenario sampled from the pool and talk with the scene characters in it. (iii) Selection: for each test character turn, the pipeline queries responses from both source and base models, with the judge model determining the evaluation dimension and selecting the superior output. All items marked ⭐ are customizable. More explanations are presented in Section 3.

on specific intended tasks. Moreover, realizing the ambitious vision of immersive virtual worlds populated with diverse intelligent NPCs (Park et al., 2023; Ran et al., 2025) requires methodological and evaluative foundations that are not only broad in scope but also flexible enough to adapt to different environments and objectives. While recent work such as Neeko (Yu et al., 2024) has made progress on the training side through dynamic LoRA to seen and unseen characters, the evaluation side remains significantly underdeveloped.

To address these needs, we propose FURINA-Builder (Figure 1), a novel pipeline for dynamically constructing RP benchmarks tailored to specific requirements. Inspired by the design of LLM-based agents (Wolter et al., 2025; Li et al., 2025a), which integrate LLMs as the core for understanding, planning, and interaction, we design a multi-agent collaboration mechanism to generate RP benchmarks with user-specified character personas, dialogue structures and evaluation dimensions. To the best of our knowledge, FURINA-Builder is the first benchmark builder for RP scenarios.

FURINA-Builder provides extensive flexibility for constructing RP benchmarks. Users can define arbitrary characters in a key–value dictionary format ⭐, specify a character–scene pool ⭐ from which RP simulations are initiated, and impose restrictions on character profile visibility ⭐ to make conversations more realistic and challenging by limiting the information available during interactions. Within each simulation, a director model determines who speaks next. When it is the test character's turn, candidate responses are generated by a source model, which serves as the primary test character driver of the simulation, and a strong baseline model, which functions as a reference to secure a performance floor. An LLM-based judge then choose one suitable evaluation dimension (e.g., coherence, factuality, faithfulness) ⭐ and compares two outputs, selecting the superior response as the final utterance. This mechanism ensures that dialogue trajectories are of high quality and that each test utterance position is assigned an appropriate evaluation criterion. By conducting simulations with different source models, users can ensure diverse data sources for their own RP benchmarks. All components marked with ⭐ are fully customizable.

Building on this pipeline, we develop FURINA-Bench, a comprehensive benchmark that integrates both established characters (with well-defined personas) and synthesized characters (newly created from scratch). Our benchmark is the first to unify these two character types in group chat scenarios and to systematically examine both reasoning and RP hallucination. This setting is crucial for simulating immersive virtual worlds, where diverse intelligent NPCs must coexist and interact. Each test utterance in group-chat settings is paired with an appropriate evaluation dimension, ensuring fine-grained assessment of models' dual capability to embody both character types effectively. To vali-

date the design, we conduct human evaluations, which confirm the builder's effectiveness and benchmark's soundness. A preliminary analysis also indicates that FURINA-Bench achieves stronger separability, clearly distinguishing model behaviors in RP scenarios. Using FURINA-Bench, we systematically evaluate a wide range of cutting-edge LLMs, including closed- and open-source, as well as "thinking" and "non-thinking" models. The results reveal several important insights. First, o3 and DeepSeek-R1 achieve the strongest overall performance on English and Chinese RP tasks, respectively. Second, across all models, established characters consistently outperform synthesized ones, reflecting the importance of dedicated training. And reasoning capabilities further amplify this gap, as they tend to weaken instruction-following abilities. Third, regarding RP hallucination, we observe no monotonic relationship between model scale and hallucination rates, indicating that factors such as training data composition may play a more critical role. Fourth, while reasoning enhances RP performance, it simultaneously increases hallucination severity through more aggressive response strategies and extended thinking processes. Finally, we identify a constraining Pareto frontier (Pareto, 1906) between RP performance and reliability, revealing an inherent trade-off in current models. In summary, our main contributions are:

- We propose **FURINA-Builder**, the first multi-agent collaboration pipeline for automatically constructing fully customizable RP benchmarks at arbitrary scales. Human evaluation justifies our builder design.

- We introduce **FURINA-Bench**, a comprehensive RP benchmark built with **FURINA-Builder**, which incorporates both established and synthesized test characters in group-chat scenarios, accompanied by fine-grained evaluation criteria. A preliminary analysis demonstrates that it facilitates clearer model separability and supports more robust evaluation.

- We conduct extensive evaluations of cutting-edge LLMs on **FURINA-Bench**, where we not only derive key insights but also provide a comprehensive analysis of their performance and limitations.

## 2 RELATED WORK

**Role-playing Benchmarks.** Recent advances in LLMs have significantly improved the capabilities of RPCAs. To evaluate these abilities, researchers have introduced a range of benchmarks targeting different aspects of RP performance. Early works target established character-level evaluation with various datasets from automatic LLM generation (Wang et al., 2023; Shao et al., 2023) to authentic data extraction and human annotation (Li et al., 2023; Chen et al., 2023; Tu et al., 2024), focusing on character fidelity and narrative grounding. Continuing along with automatic generation, RoleLLM (Wang et al., 2024) proposes a unified framework to improve models' RP performance by role construction, knowledge elicitation, and instruction tuning, accompanied by RoleBench, a fine-grained character-level benchmark with established characters. Currently, OpenCharacter (Wang et al., 2025a) evaluates the LLM role-playing capabilities of synthesized characters. Beyond general RP evaluation, several studies also focus on other aspects. RAIDEN (Wu et al., 2025a) introduces a carefully human-constructed RP dataset designed to assess one evaluation dimension per instance, effectively mitigating cross-dimension interference. RoleMRC (Lu et al., 2025) extends RP evaluation to instruction-following by combining multi-turn chats, reading comprehension, and nested tasks into a benchmark. Additionally, CoSER (Wang et al., 2025b) is the latest comprehensive authentic RP dataset with group-chat dialogues from books, but its characters are all established and it lacks fine-grained metrics for response-level evaluation. The characteristics of our benchmark compared to others are summarized in Table 1.

**LLM-based Agents.** LLM-based agents have emerged as a paradigm shift from traditional rule-based systems to sophisticated autonomous entities capable of tool use (Yao et al., 2023; Shen et al., 2023), planning (Zhang et al., 2024a; Wang & Liu, 2024; Zhang et al., 2025b), and feedback learning (Shinn et al., 2023; Zhang et al., 2025a). Early attempts like ReAct (Yao et al., 2023) introduces the synergy between reasoning and acting, enabling single agent to generate interleaved reasoning traces and task-specific actions. Recently, multi-agent systems (Hong et al., 2023; Zhang et al., 2024b; Li et al., 2025a) enhance the problem-solving capabilities of individual agents by enabling each agent to collaborate with each other and distributing the entire task to different agents, especially in data synthesis field (Tang et al., 2024a; Prabhakar et al., 2025). In our work, we apply a multi-agent collaboration approach to an automatic pipeline for RP benchmark construction.

Table 1: Comparison between FURINA-Bench and existing RP datasets. For characters, our dataset supports both established (Established) and synthesized (Synthesized) characters, and features persona changing across time (Dynamic) and subjective-objective divergence in understanding the character profile (Perspective Misalignment). For conversations, our dataset supports conversation involving more than two characters (Multi-character), character motivations and dialogue environment (Scenario), and various reply strategies of characters (Strategy).

| Dataset | Character | | | | Conversation | | | | |
|---|---|---|---|---|---|---|---|---|---|
| | Established | Synthesized | Dynamic | Perspective Misalignment | #Conv. | #Turns | Multi-character | Scenario | Strategy |
| CharacterGLM | ✓ | ✗ | ✗ | ✗ | 1,043 | 15.8 | ✗ | ✗ | ✗ |
| ChatHaruhi | ✓ | ✗ | ✗ | ✗ | 54,726 | 3.8 | ✓ | ✓ | ✗ |
| CharacterBench | ✓ | ✗ | ✗ | ✗ | 3,162 | 11.3 | ✗ | ✓ | ✗ |
| RAIDEN | ✓ | ✗ | ✗ | ✗ | 1,350 | 28.9 | ✗ | ✓ | ✗ |
| CoSER | ✓ | ✗ | ✓ | ✗ | 29,798 | 13.2 | ✓ | ✓ | ✗ |
| RoleLLM | ✓ | ✗ | ✗ | ✗ | 140,726 | 2 | ✗ | ✗ | ✗ |
| RoleMRC | ✓ | ✗ | ✗ | ✗ | 1,400 | 4 | ✗ | ✗ | ✗ |
| OpenCharacter | ✗ | ✓ | ✗ | ✗ | 10,000 | 2 | ✗ | ✓ | ✗ |
| FURINA-Bench (ours) | ✓ | ✓ | ✓ | ✓ | 1,494 | 19.8 | ✓ | ✓ | ✓ |

# 3 FURINA-BUILDER

In this section, we present a detailed introduction to FURINA-Builder (Figure 1), covering its four key components: the test character (section 3.1), the character–scene pool (section 3.2), the simulation process (section 3.3), and the selection mechanism (section 3.4). The process of constructing FURINA-Bench will be described separately in section 4.

## 3.1 TEST CHARACTER

To maximize adaptability across diverse role-playing tasks, FURINA-Builder imposes minimal constraints on test character inputs. A test character is defined as $c_{\text{test}} = (k_i, v_i, \tau_i)_{i=1}^n$, where the profile consists of key–value pairs $(k_i, v_i)$, each annotated with a visibility label $\tau_i \in \{\text{public}, \text{private}\}$. Users may designate certain attributes as private, and the pipeline ensures these remain hidden from other characters during interaction, thereby enhancing realism. Beyond user-defined inputs, we also provide a standardized pipeline for synthesizing characters, enabling efficient construction of task-specific personas. Full details of this pipeline are given in Appendix A.

## 3.2 CHARACTER-SCENE POOL

The character–scene pool $\mathcal{S} = \{s_1, s_2, \ldots, s_{|\mathcal{S}|}\}$ serves as a large-scale repository of high-quality dialogue scenarios for FURINA-Builder. Dialogues situated within structured scenarios are more effective for evaluating RP capabilities than free-form chat. Given a test character $c_{\text{test}}$, the pipeline samples scenarios $s \in \mathcal{S}$ and inserts $c_{\text{test}}$ into $s$ to interact with scene characters. Each scenario $s = \langle B, M, D_{\text{orig}}, \mathcal{C}_{\text{scene}} \rangle$ consists of four components: (i) background $B$ (world setting and current environment), (ii) motivation $M$ (character purposes and situational drivers), (iii) original dialogue $D_{\text{orig}}$ (authentic scripts from literary works used as references), and (iv) scene characters $\mathcal{C}_{\text{scene}}$ drawn from those scripts. Users may further extend the pool with custom scenarios, i.e., $\mathcal{S} \leftarrow \mathcal{S} \cup s_{\text{custom}}$. To build the pool, we curated 6,556 scenario fragments from a bilingual corpus $\mathcal{B} = \mathcal{B}_{\text{zh}} \cup \mathcal{B}_{\text{en}}$ consisting of 80 Chinese and 100 English books, where character profiles adapt to temporal context so that the same character can have different profiles. This ensures that test characters can be evaluated across a diverse set of ready-to-use scenarios. The full construction process is detailed in Appendix B.

## 3.3 SIMULATION

During each dialogue simulation $s$, a director model $\mathcal{M}_{\text{director}}$ iteratively determines the next speaker or decides whether to terminate the conversation, and the complete character set is $\mathcal{C}_{\text{total}} = \{c_{\text{test}}\} \cup \mathcal{C}_{\text{scene}}$. The reply prompt for the test character is defined as $\text{Prompt}(c_{\text{test}}) = \langle B, c_{\text{test}}, \mathcal{P}_{\text{pub}}(\mathcal{C}_{\text{scene}}), H \rangle$, where $\mathcal{P}_{\text{pub}}(\mathcal{C}_{\text{scene}})$ extracts the public attributes of all scene characters $\mathcal{C}_{\text{scene}}$ and $H$ denotes dialogue history. For each scene character $c_i \in \mathcal{C}_{\text{scene}}$ role-played by the model $\mathcal{M}_{\text{scene}}$, the prompt is further enriched as $\text{Prompt}(c_i) = \langle B, c_i, \mathcal{P}_{\text{pub}}(\mathcal{C}_{\text{total}} \setminus \{c_i\}), H, M, D_{\text{orig}} \rangle$, which additionally incorporates the motivation $M$ and the original dialogue references $D_{\text{orig}}$. To ensure balanced coverage across evaluation dimensions $\mathcal{D}_{\text{eval}} = \{d_1, d_2, \ldots, d_{|\mathcal{D}_{\text{eval}}|}\}$ (detailed in

Table 2: Accuracy of the dimension selection using GPT-4.1 judge.

| Dimension | #CR | #FR | #RR | #CA | #PA | Average Score |
|---|---|---|---|---|---|---|
| Accuracy | 0.909 | 0.862 | 0.891 | 0.908 | 0.888 | 0.892 |

Table 3: Pearson correlations and p-values between model scores and human annotations.

| Metric | GPT-4.1 | DeepSeek-R1-0528 | DeepSeek-V3-0324 |
|---|---|---|---|
| CR | **0.7075(0.0000)** | 0.6285(0.0000) | 0.4875(0.0024) |
| FR | **0.6630(0.0000)** | 0.5788(0.0002) | 0.5995(0.0001) |
| RR | 0.5602(0.0002) | **0.6304(0.0000)** | 0.5937(0.0001) |
| CA | **0.6174(0.0000)** | 0.6103(0.0000) | 0.4296(0.0044) |
| PA | **0.5877(0.0002)** | 0.4677(0.0073) | 0.4346(0.0153) |
| Average | **0.6275(0.0000)** | 0.5808(0.0000) | 0.4982(0.0000) |

Section 4.1), alongside normal conversation, we employ a targeted dimension-emphasis mechanism with *Dynamically Weighted Random Selection* algorithm, which adaptively adjusts occurrence probabilities to emphasize underrepresented dimensions. Full details are provided in Appendix C.

## 3.4 SELECTION

For every turn of the test character $c_{\text{test}}$, FURINA-Builder queries both the source model $\mathcal{M}_{\text{source}}$ and base model $\mathcal{M}_{\text{base}}$ to generate candidate responses $r_{\text{source}}$ and $r_{\text{base}}$, respectively. A judge model $\mathcal{M}_{\text{judge}}$ then selects the most appropriate evaluation dimension $d^* \in \mathcal{D}_{\text{eval}}$ given the current context. Based on $d^*$, the judge performs chain-of-thought (CoT) reasoning (Wei et al., 2022) to rate the candidate outputs on a 5-point Likert scale (Likert, 1932), yielding a score $\sigma \in \{1, 2, 3, 4, 5\}$, where 1 indicates strong preference for $\mathcal{M}_{\text{source}}$ and 5 indicates strong preference for $\mathcal{M}_{\text{base}}$. The dialogue history is updated accordingly:

$$H_{t+1} = \begin{cases} H_t \cup \{r_{\text{source}}\} & \text{if } \sigma \leq 3 \\ H_t \cup \{r_{\text{base}}\} & \text{if } \sigma > 3 \end{cases} \tag{1}$$

This mechanism ensures that weaker responses are filtered out, maintaining coherent and high-quality dialogue trajectories. Otherwise, accumulated inconsistencies in the dialogue history could distort the context such that the good response becomes trivial (e.g., clarifying confusion), hindering our ability to assess RP ability itself. Unlike prior evaluation methods (Tu et al., 2024; Wang et al., 2025b), which score all dimensions simultaneously, our approach evaluates along a single targeted dimension, thereby reducing cross-dimension interference. Importantly, because not every utterance naturally reflects performance across all dimensions, forcing simultaneous judgments may introduce noise and hinder fine-grained evaluation, as mentioned in Wu et al. (2025a). Once the dimension $d^*$ is selected, we fix the test utterance and dimension pairing $\langle u_{\text{test}}, d^* \rangle$ for benchmark construction.

## 4 FURINA-BENCH

In this section, we introduce the configuration of FURINA-Builder (section 4.1) and the building pipeline of FURINA-Bench (section 4.2), followed by dataset statistics (section 4.3). We then validate the reliability and robustness of the pipeline and benchmark design (section 4.4) prior to the evaluation analysis.

## 4.1 CONFIGURATION

**Model Selection.** For FURINA-Bench, we adopt a diverse set of cutting-edge LLMs as source models $\mathcal{M}_{\text{source}}$, covering both general and reasoning LLM series. Our selection spans from mid-scale open-source models (e.g., 24B–70B) to large frontier proprietary systems (e.g., GPT, Claude, Gemini series), ensuring a broad and representative dataset foundation. We further employ Qwen3-235B-A22B for the scene character model $\mathcal{M}_{\text{scene}}$ and director model $\mathcal{M}_{\text{director}}$, and GPT-4.1 for both the base model $\mathcal{M}_{\text{base}}$ and the judge model $\mathcal{M}_{\text{judge}}$. The complete list of models, together with their sources and version details, is provided in Appendix D.

Table 4: The performance of various LLMs on FURINA-Bench. **Bold** and underlined values indicate highest and second-highest score, respectively. 95% confidence intervals (CI) are computed by bootstrapping with 1000 resamples. For Chinese part we exclude Llama3.1-8B, 70B and CoSER-70B models due to their poor English performance and limited Chinese training. In addition, we also exclude the Genimi models due to their red-teaming behavior and report in Appendix N.

| Model | Evaluation Dimensions | | | | | Weighted Average Score [95% CI] |
|---|---|---|---|---|---|---|
| | Context Reliance | Factual Recall | Reflective Reasoning | Conversational Ability | Preference Alignment | |
| *English Part* | | | | | | |
| Humanish-Llama-3.1-8B | 22.70 | 16.20 | 21.08 | 22.00 | 33.76 | 23.15[−0.0090, 0.0095] |
| Peach-2.0-9B-8k-Roleplay | 12.40 | 7.56 | 6.50 | 13.09 | 14.29 | 10.77[−0.0066, 0.0069] |
| Llama3.1-8B | 10.85 | 11.92 | 10.12 | 8.63 | 8.42 | 9.99[−0.0076, 0.0077] |
| Qwen3-8B | 12.78 | 12.00 | 7.53 | 10.55 | 24.07 | 13.39[−0.0087, 0.0092] |
| Qwen3-8B-thinking | 14.28 | 13.39 | 7.67 | 11.20 | 23.27 | 13.96[−0.0088, 0.0088] |
| Qwen3-32B | 15.98 | 13.44 | 9.08 | 15.29 | 20.21 | 14.80[−0.0088, 0.0089] |
| Qwen3-32B-thinking | 28.49 | 27.47 | 13.90 | 27.72 | 38.78 | 27.27[−0.0117, 0.0117] |
| Llama3.1-70B | 14.62 | 12.01 | 11.74 | 12.48 | 10.84 | 12.34[−0.0080, 0.0081] |
| CoSER-70B | 11.16 | 10.42 | 5.89 | 9.89 | 9.90 | 9.45[−0.0077, 0.0077] |
| Qwen3-235B | 19.61 | 18.99 | 19.88 | 23.16 | 32.23 | 22.77[−0.0105, 0.0102] |
| Qwen3-235B-thinking | 27.37 | 35.45 | 21.88 | 28.08 | 45.50 | 31.66[−0.0124, 0.0125] |
| GPT-4o | 25.70 | 22.94 | 19.53 | 26.57 | 28.70 | 24.69[−0.0080, 0.0081] |
| Deepseek-V3 | 22.51 | 21.27 | 14.44 | 19.51 | 29.87 | 21.52[−0.0096, 0.0096] |
| Deepseek-R1 | 36.31 | **36.18** | 33.39 | 32.50 | 44.14 | 36.50[−0.0130, 0.0132] |
| Claude-4-Sonnet | 32.62 | 27.98 | 48.30 | 30.41 | 41.74 | 36.21[−0.0112, 0.0112] |
| Claude-4-Sonnet-thinking | 30.80 | 29.72 | **53.45** | 27.71 | 40.90 | 36.52[−0.0113, 0.0114] |
| o3 | **42.99** | 35.31 | 43.00 | **42.68** | **55.93** | **43.98**[−0.0122, 0.0118] |
| *Chinese Part* | | | | | | |
| Chatglm3-6B | 13.53 | 12.38 | 15.35 | 12.60 | 13.88 | 12.96[−0.0025, 0.0026] |
| Peach-2.0-9B-8k-Roleplay | 15.31 | 11.45 | 9.55 | 19.63 | 28.08 | 16.93[−0.0052, 0.0077] |
| Qwen3-8B | 45.07 | 40.32 | 43.24 | 39.81 | 59.69 | 45.63[−0.0109, 0.0053] |
| Qwen3-8B-thinking | 53.23 | 49.06 | 41.21 | 49.17 | 69.12 | 52.36[−0.0115, 0.0116] |
| Qwen3-32B | 64.14 | 57.43 | 55.11 | 60.49 | 78.34 | 63.10[−0.0105, 0.0106] |
| Qwen3-32B-thinking | **71.39** | 68.42 | 54.90 | 67.84 | **82.71** | 69.05[−0.0108, 0.0107] |
| Qwen3-235B | 59.38 | 54.30 | 56.87 | 56.38 | 72.92 | 59.97[−0.0109, 0.0111] |
| Qwen3-235B-thinking | 67.07 | **70.55** | 57.68 | **70.00** | 81.39 | 69.34[−0.0107, 0.0103] |
| GPT-4o | 28.49 | 19.78 | 21.68 | 23.67 | 28.94 | 24.51[−0.0071, 0.0073] |
| Deepseek-V3 | 36.31 | 44.25 | 21.49 | 31.84 | 49.95 | 36.77[−0.0106, 0.0106] |
| Deepseek-R1 | 67.91 | 70.51 | **80.19** | 65.77 | 82.53 | **73.38**[−0.0097, 0.0097] |
| Claude-4-Sonnet | 37.77 | 32.48 | 65.24 | 35.48 | 49.10 | 44.02[−0.0104, 0.0103] |
| Claude-4-Sonnet-thinking | 40.17 | 37.62 | 69.75 | 36.57 | 48.56 | 46.33[−0.0104, 0.0106] |
| o3 | 47.46 | 39.06 | 54.60 | 47.86 | 56.77 | 50.15[−0.0106, 0.0106] |

**Evaluation Dimension Definitions.** Based on prior role-playing studies, we include five key dimensions $\mathcal{D}_{eval} = \{d_1, d_2, d_3, d_4, d_5\}$ in our evaluation framework: (1) *Context Reliance (CR)* measures appropriate utilization of contextual information; (2) *Factual Recall (FR)* evaluates application of general world knowledge which is not explicitly provided; (3) *Reflective Reasoning (RR)* assesses human-like reasoning and justification abilities; (4) *Conversational Ability (CA)* examines multi-turn dialogue management and coherent conversation flow; (5) *Preference Alignment (PA)* measures single-turn response quality by penalizing robotic or repetitive responses. Detailed definitions and relevant prompts of each dimension used in construction are provided in Appendix E.

**Test Characters.** Our experiments involve 20 representative test characters, evenly distributed across four categories: 5 synthesized Chinese, 5 synthesized English, 5 established Chinese, and 5 established English characters. Such a balanced distribution across language and role types strengthens the benchmark's generalizability. The profiles of test characters are provided in Appendix F.

## 4.2 BUILDING PIPELINE

The benchmark is constructed under the configuration specified above. Given the large number of source models, we set a minimum evaluation threshold of $\tau = 10$, meaning that the pipeline continues until each character reaches this threshold for all evaluation dimensions. To ensure data quality, we also employ both LLM-based and rule-based filtering methods, and more details can be found in Appendix G. Furthermore, a case study is presented in Appendix H.

Table 5: The performance of established & synthesized characters on FURINA-Bench. **Bold** and underlined values indicate highest and second-highest score, respectively. Gap presents the disparity between established and synthesized characters. Evaluation dimension details are in Appendix Q.

| Model | Test Characters | | | | | |
| --- | --- | --- | --- | --- | --- | --- |
| | English Part | | | Chinese Part | | |
| | Established | Synthesized | Gap | Established | Synthesized | Gap |
| Peach-2.0-9B-8k-Roleplay | 9.84 | 11.89 | -2.05 | 13.91 | 16.65 | -2.74 |
| Qwen3-8B | 13.46 | 13.42 | +0.04 | 46.96 | 44.03 | +2.93 |
| Qwen3-8B-thinking | 17.04 | 10.52 | +6.52 | 59.22 | 44.09 | +15.13 |
| Qwen3-32B | 16.78 | 12.42 | +4.36 | 67.05 | 58.35 | +8.7 |
| Qwen3-32B-thinking | 32.79 | 21.07 | +11.92 | 76.85 | 59.61 | +17.24 |
| Qwen3-235B | 24.40 | 20.93 | +3.47 | 63.74 | 55.50 | +8.24 |
| Qwen3-235B-thinking | 37.55 | 25.12 | +12.43 | 76.35 | 60.86 | +15.49 |
| GPT-4o | 25.32 | 24.05 | +1.27 | 26.60 | 21.96 | +4.65 |
| Deepseek-V3 | 24.21 | 18.29 | +5.92 | 43.93 | 28.03 | +15.9 |
| Deepseek-R1 | 45.46 | 26.44 | +19.02 | **77.16** | **68.85** | +8.31 |
| Claude-4-Sonnet | 40.75 | 31.17 | +9.58 | 48.67 | 38.36 | +10.31 |
| Claude-4-Sonnet-thinking | 41.95 | 30.53 | +11.42 | 50.65 | 41.00 | +9.65 |
| o3 | **49.94** | **37.39** | +12.55 | 55.66 | 43.42 | +12.24 |

## 4.3 DATASET STATISTICS

FURINA-Bench is a bilingual role-playing dataset consisting of 20 test characters, evenly split between Chinese and English personas, encompassing both established and synthesized identities. These characters interact with 1,471 unique roles, producing 1,459 high-quality multi-party dialogues with 7,181 test utterances. All five evaluation dimensions are balanced across languages, each containing over 500 examples. We summarize the dataset statistics and dimension distributions in Appendix I. The dataset is derived from various cutting-edge source models, with their distributions detailed in Appendix J.

## 4.4 BENCHMARK DESIGN VALIDATION

We begin by validating the key components of FURINA-Builder, including dimension selection and judgment scoring. To this end, five annotators were recruited, with detailed guidelines provided in Appendix K and Appendix L. The annotation interface is shown in Appendix M. We further conduct a preliminary analysis of model separability on FURINA-Bench to demonstrate its robustness. Beyond this benchmark, the builder's effectiveness is reflected in its ability to handle diverse character settings and evaluation dimensions in a consistent and human-aligned manner.

**Reliable Dimension Selection.** During construction, we employ GPT-4.1 to identify the appropriate evaluation dimension. On 1000 manually annotated samples, it achieves an average accuracy of 0.892 with stable performance across all five dimensions (Table 2), demonstrating high reliability.

**Good Dimension-based Judgment.** We further examine the consistency between model-based and human scoring. On 400 samples, the Pearson correlation between GPT-4.1 and human judgments is high, exceeding that of alternatives such as DeepSeek-R1-0528 and DeepSeek-V3-0324 (Table 3). These results indicate that GPT-4.1 aligns closely with humans in dimension-specific scoring.

**Better Separability.** A robust benchmark should be separable, clearly differentiating models of varying capabilities (Li et al., 2024). We compare our evaluation method (introduced in section 5.1) against the GCA baseline (Wang et al., 2025b). As shown in Figure 3, FURINA-Bench yields a steeper slope, demonstrating stronger discriminative power with more challenging evaluation settings in RP scenarios. More details and quantitative comparison are shown in Appendix O.

## 5 BENCHMARK EVALUATION

### 5.1 EVALUATION METHOD

The FURINA-Bench dataset is a curated dialogue corpus $\mathcal{D}_{\text{bench}} = \{(H_i, u_i, d_i)\}_{i=1}^{N}$, where each instance consists of a simulated dialogue history $H_i$, a corresponding test utterance $u_i$, and a pre-assigned evaluation dimension $d_i \in \mathcal{D}_{\text{eval}}$. For evaluation, the test model $\mathcal{M}_{\text{test}}$ and base model $\mathcal{M}_{\text{base}}$ generate responses $r_{\text{test},i}$ and $r_{\text{base},i}$ respectively using PromptEval: PromptEval($c_{\text{test}}$) =

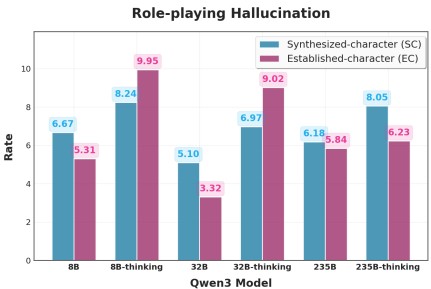

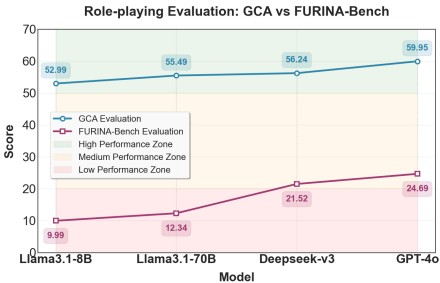

Figure 2: Role-playing hallucination rates (%) of Qwen3 with Synthesized-character and Established-character on Chinese section. Reasoning produces more serious hallucination.

Figure 3: Role-playing evaluation results across four models using GCA Evaluation and our FURINA-Bench Evaluation. Our method illustrates more challenging with better separability.

$\text{Prompt}(c_{\text{test}}) \cup \mathcal{S}(d_i)$, where $\text{Prompt}(c_{\text{test}})$ is defined in section 3.3 and $\mathcal{S}(d_i)$ is the response strategy aligned with the pre-assigned dimension. Since response strategies are widely adopted in RP applications, this design not only enhances the realism of the evaluation but also improves scoring efficiency. A clear evaluation diagram is shown in Appendix P. Furthermore, to mitigate ordering bias in LLM-as-a-judge evaluation, comparisons are conducted bidirectionally. The judge model $\mathcal{M}_{\text{judge}}$ scores both orders $(r_{\text{test},i}, r_{\text{base},i})$ and $(r_{\text{base},i}, r_{\text{test},i})$, producing $\sigma_{i,1}$ and $\sigma_{i,2}$ on a 5-point Likert scale. The final comparative score for instance $i$ is computed as:

$$\text{Score}_i = \frac{1}{2}[f(\sigma_{i,1}) + f(6 - \sigma_{i,2})] \tag{2}$$

where $f : \{1, 2, 3, 4, 5\} \to \{3, 1, 0.5, 0, 0\}$ is the unbalanced scoring function. The test response can receive a score only if it meets or exceeds the base response, penalizing any performance below the reference baseline while providing enhanced rewards for exceptionally high-quality responses.

Overall performance is defined as the normalized score ratio, calculated as the total score obtained divided by the maximum possible score:

$$\text{Performance} = \frac{\sum_{i=1}^{N} \text{Score}_i}{\sum_{i=1}^{N} \max(f(\sigma))} = \frac{\sum_{i=1}^{N} \text{Score}_i}{3N} \tag{3}$$

We continue to use GPT4.1 as both the base and judge model following a set of initial explorations. The complete set of response strategy prompts and a detailed case study of FURINA-Bench evaluation are provided in Appendix R and S, respectively.

## 5.2 MAIN RESULTS

**Our benchmark uncovers clear role-playing performance patterns across languages and model families.** We evaluate various LLMs and table 4 summarizes the main findings: *1)* For English texts, o3 achieves leading scores in Context Reliance, Conversational Ability, and Preference Alignment, yielding the highest overall RP score (43.98) and confirming its dominance in English RP scenarios. In contrast, Chinese performance is more distributed: Qwen3-32B/235B-thinking perform best across most RP dimensions, while DeepSeek-R1 achieves the strongest overall RP score (73.38). *2)* Qwen3 series exhibits clear scaling benefits for all dimensions. Performance increases steadily from 8B (13.39/45.63) to 32B (14.80/63.10) and to 235B-A22B (22.77/59.97) parameters for English and Chinese, respectively. *3)* Qwen3 models deliver significantly stronger results in Chinese, even at smaller scales, emphasizing the importance of language-specific training data in developing effective RPCAs. *4)* The choice of the base model plays a critical role. GPT-4.1, used as the English baseline, is stronger than the Chinese baseline, which depresses English test scores overall and underscores the importance of base model selection for FURINA-Builder. *5)* At comparable model sizes, dedicated RP models generally outperform general LLMs, particularly in emotional intelligence during dialogue (CA) and human-like logical reflection (RR). In Chinese, Qwen3-8B shows impressively strong RP performance, potentially because the Qwen series has explicitly optimized role-playing as a core capability since Qwen2.5 (Qwen et al., 2025). *6)* Across the whole leaderboard, dedicated RP models remain substantially weaker than the top performers. This gap

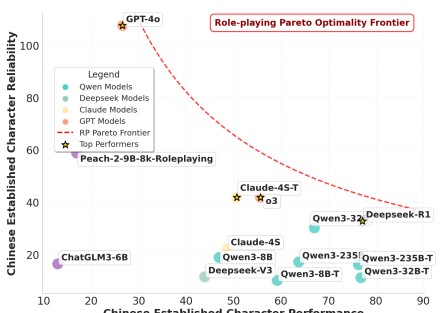 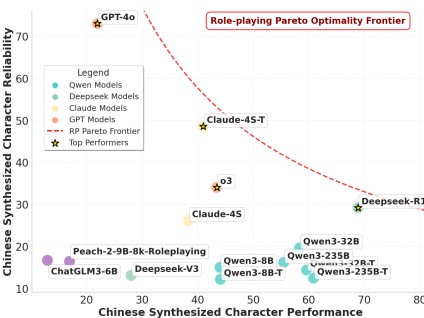

Figure 4: Relationship between role-playing performance and reliability for Chinese established (left) and synthesized (right) characters. Reliability score is computed by 100 / (hallucination rate).

largely reflects the much larger scale and overall intelligence of these leading models, the fact that role-playing has become an explicit optimization target in modern LLM training, and the stronger generalization abilities that large models have.

## 5.3 ANALYSIS ON REASONING UTILITY

**Reasoning generally improves model performance on RP tasks, but the mechanisms of improvement differ across models.** For Qwen3, reasoning serves as a robust enhancement, improving performance across multiple dimensions including contextual consistency (CR), fundamental world knowledge (FR), human-like reflection (RR), conversational intelligence (CA), and dialogue tone (PA). For Claude-4-Sonnet, however, the improvements from reasoning are predominantly concentrated in the FR and RR dimensions, while performance in the remaining dimensions even shows signs of decline. This discrepancy is likely attributable to the distinct reasoning training approaches, where more generalized reasoning algorithms or exposure to RP-related data during reasoning training tend to yield more reliable improvements in RP tasks.

## 5.4 ESTABLISHED AND SYNTHESIZED CHARACTERS

**Established characters consistently outperform synthesized characters, and reasoning capabilities further amplify this discrepancy**. In addition to the overall performance, we analyze model RP capabilities across different character types. As illustrated in Table 5, all models consistently perform better on established characters than synthesized characters, with performance gaps ranging from minimal (+0.04 for Qwen3-8B in English) to substantial (+19.02 for DeepSeek-R1 in English). Established characters primarily reflect character understanding grounded in model training, whereas synthesized characters rely more on in-context learning abilities. Their performance disparities thus highlight the necessity of dedicated training for advancing RP capabilities. Additionally, we observe that reasoning models demonstrate significantly larger gaps, suggesting that reasoning mechanisms preferentially leverage pre-existing character knowledge over prompt-provided character details. This finding can be attributed to the fact that synthesized character performance significantly reflects instruction-following capabilities, but current reasoning mechanisms weaken this proficiency, which is also discussed in recent works (Li et al., 2025b; Wu et al., 2025b).

## 5.5 ROLE-PLAYING HALLUCINATION

Hallucination is a critical challenge in RP, as it can significantly undermine the immersive experience. Prior work, such as Tang et al. (2024b), broadly defines RP hallucination as any deviation from predefined character roles or responses that are inconsistent with the intended persona. However, this definition is too general and limits fine-grained analysis. Following the taxonomy of Huang et al. (2025), hallucinations can be divided into two categories: *factuality hallucinations*, where outputs contradict real-world facts, and *faithfulness hallucinations*, where outputs deviate from given instructions or context. Building on this categorization, we refine RP hallucinations into two subtypes. *Established-character (EC) hallucination*, a form of *factuality hallucination*, occurs when models misrepresent knowledge already encoded in pretraining and are assessed through test utterances tagged with Factual Recall. In contrast, *Synthesized-character (SC) hallucination*, corresponding to *faithfulness hallucinations*, arises when models fail to remain consistent with context

information, and are captured through utterances tagged with Context Reliance. Furthermore, to more accurately measure RP hallucination rates, we employ an automatic checker $\mathcal{M}_{\text{checker}}$ (e.g., Qwen2.5-32B-Instruct) to detect hallucination-related keywords in CoT judgments generated during evaluation. We then compute their occurrence probability as a more precise and reliable metric of RP hallucination. The detailed checker prompt is provided in Appendix T.

**Reasoning improves RP performance but amplifies hallucinations.** As shown in Figure 2, all Qwen3 variants exhibit higher hallucination rates in thinking mode compared to no-thinking mode, for both SC and EC. While reasoning boosts performance, aggressive response strategies and extended thinking significantly increase hallucinations. Notably, model scale does not follow a monotonic trend with hallucination rates, suggesting that RP hallucinations arise primarily from training data rather than from inherent capacity limitations. The impact of reasoning is especially evident in the 8B and 32B models, where SC hallucinations increase sharply, highlighting challenges in effectively leveraging information provided in prompts. Full statistics are reported in Appendix U.

**A Pareto frontier emerges between RP performance and reliability.** Figure 4 illustrates the trade-off between RP performance and reliability, where reliability is defined as the inverse hallucination rate. Our analysis shows that high-performing models (e.g., Deepseek-R1) often compromise reliability, while highly reliable models tend to adopt conservative strategies that suppress RP performance (e.g., GPT-4o). The Qwen3 series demonstrates strong RP capabilities but limited reliability, whereas Claude-4-Sonnet in thinking mode achieves the best balance among all evaluated models. This trade-off highlights a core challenge for current LLMs: How to achieve simultaneous gains in both RP performance and reliability to surpass the existing Pareto frontier. We suggest that advances in pre-training data curation, post-training with diverse role types, and instruction-sensitive reasoning mechanisms represent promising directions toward resolving this challenge. Additional results for English tasks are reported in Appendix V. Furthermore, as shown in Appendix U, for dedicated RP models, they tend to exhibit lower RP hallucination rates than general LLMs of similar scale. For example, Humanish-Roleplay-Llama-3.1-8B on the English Synthesized Character even outperforms the top performers. We believe this is primarily due to overfitting on the RP task, and there is no explicit risk of cross-domain goal conflict.

## 6 CONCLUSION

In this work, we present FURINA-Builder, a novel multi-agent collaboration pipeline for constructing fully customizable role-playing benchmarks at arbitrary scales. Using this pipeline, we build FURINA-Bench, a comprehensive benchmark including both established and synthesized characters with fine-grained evaluation criteria. We further verify the reliability and robustness of the pipeline and the benchmark design. Our evaluation across a wide range of cutting-edge LLMs shows that o3 and DeepSeek-R1 achieve the strongest performance on English and Chinese RP tasks, respectively. We find that established characters consistently outperform synthesized ones, and that reasoning capabilities, while enhancing RP performance, also amplify hallucination risks. Beyond these findings, we reveal a more general Pareto frontier between RP performance and reliability across most models. These results underscore the effectiveness of FURINA-Builder and the challenging nature of FURINA-Bench, establishing a solid foundation for future RP evaluation research.

## 7 ETHICS STATEMENT

We officially employed five professional annotators from technology companies, each with over one year of role-playing annotation experience in gaming areas, ensuring quality and expertise. All of them were compensated at levels exceeding typical market rates, in accordance with fair-labor and ethical research standards. We will release our pipeline and benchmark to promote progress in RPLA evaluation. We clarify that the provided well-constructed character–scene pool is solely intended for research purposes, and any commercial user is expected to construct their own pools tailored to their specific scenarios, which aligns with our original motivation of enabling customizable benchmark construction. In addition, we strictly adhere to copyright policies and do not distribute any raw novel content. All data in our dataset has been adjusted and transformed by LLMs and is intended solely for research purposes, and we require that users of our work obtain proper permissions when creating derivative resources. As FURINA-Bench is derived from literary material, the content presented may be not reflect the views of the original authors. Finally, we acknowledge potential risks and encourage responsible, research-focused use of our benchmark.

## 8 REPRODUCIBILITY STATEMENT

To ensure reproducibility of our results, we put our complete source codes into the supplementary materials. All model names and version numbers used in our experiments are detailed in the Appendix. For model inference, we employ official default parameters for closed-source models and Hugging Face recommended parameters for open-source models. Complete prompt templates and configurations are also provided in the Appendix. These resources collectively enable full replication of our experimental setup and results.

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

## A  SYNTHESIZING CHARACTERS PIPELINE

Synthetic characters are essential for constructing robust test sets to evaluate RP capabilities. In contrast to real-world or widely recognized fictional characters, which may introduce biases due to models' prior exposure during training, synthetic characters enable a more controlled and faithful assessment of a model's ability to embody and consistently sustain a novel persona. Nevertheless, designing high-quality synthetic characters remains a challenging task.

First, synthetic characters must be entirely fictional and deliberately constructed to minimize overlap with real-world public figures or established literary personas. This design prevents model responses from relying on memorized knowledge, instead grounding them in the character's defined attributes and narrative context. Second, characters should not be developed in isolation. To achieve narrative coherence, depth, and behavioral consistency, a comprehensive character profile must encompass not only surface-level traits but also internal elements such as personal memories, formative experiences, interpersonal relationships, and even a coherent worldview.

To address these challenges, we design and implement a structured, LLM-based pipeline for generating high-quality, original synthetic characters. The proposed pipeline systematically constructs multi-dimensional character profiles that support rich and consistent RP behavior. An overview of the pipeline is illustrated in Figure 5.

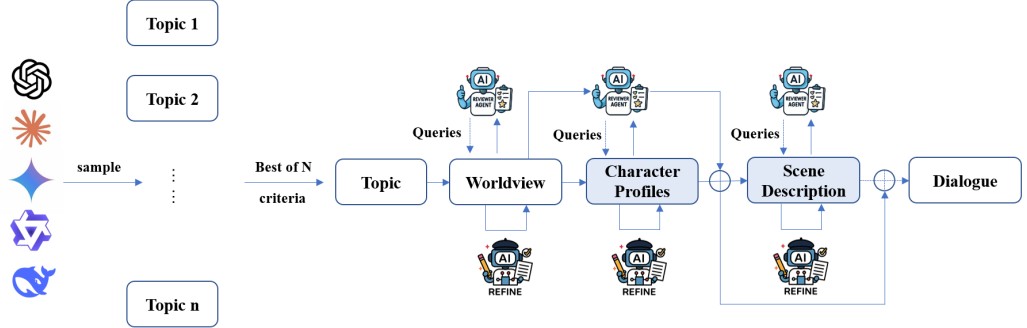

Figure 5: LLM-based synthesizing characters pipeline. Several mainstream LLMs are first employed to sample diverse thematic seeds, from which a coherent fictional worldview is constructed. Based on this worldview, multiple original character profiles are generated. A narrative scene is then created involving the selected characters, culminating in a dialogue that reflects their roles and relationships. Each intermediate step undergoes iterative self-review and refinement to ensure logical consistency, narrative coherence, and linguistic fluency.

## B  CHARACTER-SCENE POOL CONSTRUCTION

In this section, we present the data source (B.1) and the construction pipeline (B.2) in detail for the character-scene pool, and finally show one example from our well-constructed pool (B.3). We modify the book extraction codes from Wang et al. (2025b) to support the pool building.

---

[3]https://www.goodreads.com/list/show/1.Best_Books_Ever
[4]https://m.douban.com/subject_collection/ECKM5FBEI

Table 6: The 100 selected English books from Goodreads' *Best Books Ever* list[3].

| English Books | |
|---|---|
| **1**. *The Hunger Games (The Hunger Games, #1)* | **2**. *Harry Potter and the Order of the Phoenix (H. P., #5)* |
| **3**. *Pride and Prejudice* | **4**. *To Kill a Mockingbird* |
| **5**. *The Book Thief* | **6**. *Animal Farm* |
| **7**. *The Chronicles of Narnia (#1-7)* | **8**. *The Fault in Our Stars* |
| **9**. *The Picture of Dorian Gray* | **10**. *Wuthering Heights* |
| **11**. *Gone with the Wind* | **12**. *The Perks of Being a Wallflower* |
| **13**. *The Lightning Thief (Percy Jackson and the Olympians, #1)* | **14**. *The Little Prince* |
| **15**. *The Great Gatsby* | **16**. *Crime and Punishment* |
| **17**. *Memoirs of a Geisha* | **18**. *Les Misérables* |
| **19**. *The Alchemist* | **20**. *Lord of the Flies* |
| **21**. *The Hitchhiker's Guide to the Galaxy (#1)* | **22**. *The Help* |
| **23**. *Dracula* | **24**. *Ender's Game (Ender's Saga, #1)* |
| **25**. *Of Mice and Men* | **26**. *One Hundred Years of Solitude* |
| **27**. *Brave New World* | **28**. *A Thousand Splendid Suns* |
| **29**. *The Time Traveler's Wife* | **30**. *The Princess Bride* |
| **31**. *The Secret Garden* | **32**. *The Outsiders* |
| **33**. *A Game of Thrones (A Song of Ice and Fire, #1)* | **34**. *Little Women* |
| **35**. *A Wrinkle in Time (Time Quintet, #1)* | **36**. *The Odyssey* |
| **37**. *Harry Potter and the Deathly Hallows (H. P., #7)* | **38**. *Frankenstein: The 1818 Text* |
| **39**. *The Kite Runner* | **40**. *The Handmaid's Tale (The Handmaid's Tale, #1)* |
| **41**. *The Lovely Bones* | **42**. *The Adventures of Huckleberry Finn* |
| **43**. *Life of Pi* | **44**. *A Tale of Two Cities* |
| **45**. *Dune (Dune, #1)* | **46**. *Harry Potter and the Prisoner of Azkaban (H.P.,#3)* |
| **47**. *Water for Elephants* | **48**. *Harry Potter and the Sorcerer's Stone (H. P., #1)* |
| **49**. *The Bell Jar* | **50**. *Matilda* |
| **51**. *The Stand* | **52**. *Catch-22* |
| **53**. *The Adventures of Sherlock Holmes (S. H., #3)* | **54**. *The Pillars of the Earth (Kingsbridge, #1)* |
| **55**. *Rebecca* | **56**. *Great Expectations* |
| **57**. *The Girl with the Dragon Tattoo (Millennium, #1)* | **58**. *The Color Purple* |
| **59**. *Anna Karenina* | **60**. *My Sister's Keeper* |
| **61**. *The Brothers Karamazov* | **62**. *A Clockwork Orange* |
| **63**. *And Then There Were None* | **64**. *The Road* |
| **65**. *To Kill a Mockingbird* | **66**. *The Golden Compass (His Dark Materials, #1)* |
| **67**. *Vampire Academy (Vampire Academy, #1)* | **68**. *Siddhartha* |
| **69**. *The Complete Stories and Poems* | **70**. *Interview with the Vampire (The Vampire Chronicles, #1)* |
| **71**. *Don Quixote* | **72**. *The Old Man and the Sea* |
| **73**. *The Poisonwood Bible* | **74**. *Harry Potter and the Goblet of Fire (H. P., #4)* |
| **75**. *Atlas Shrugged* | **76**. *The Notebook (The Notebook, #1)* |
| **77**. *Harry Potter and the Half-Blood Prince (H. P., #6)* | **78**. *Moby-Dick or, The Whale* |
| **79**. *A Prayer for Owen Meany* | **80**. *Clockwork Angel (The Infernal Devices, #1)* |
| **81**. *The Stranger* | **82**. *The Secret Life of Bees* |
| **83**. *Harry Potter and the Chamber of Secrets (H. P., #2)* | **84**. *The Red Tent* |
| **85**. *The Name of the Wind(The Kingkiller Chronicle,#1)* | **86**. *The Master and Margarita* |
| **87**. *The Metamorphosis* | **88**. *Eragon (The Inheritance Cycle, #1)* |
| **89**. *The Count of Monte Cristo* | **90**. *Looking for Alaska* |
| **91**. *The Adventures of Tom Sawyer* | **92**. *Charlie and the Chocolate Factory(Charlie Bucket,#1)* |
| **93**. *The Last Olympian (Percy Jackson and the Olympians, #5)* | **94**. *The Curious Incident of the Dog in the Night-Time* |
| **95**. *The Shadow of the Wind (Cemetery of Forgotten Books, #1)* | **96**. *The Unbearable Lightness of Being* |
| **97**. *On the Road* | **98**. *The Name of the Rose* |
| **99**. *A Story of Yesterday* | **100**. *The Godfather (The Godfather, #1)* |

Table 7: The 80 selected Chinese books mainly from *Douban Chinese Novels TOP100* list[4].

| Chinese Books | |
|---|---|
| **1**. 一个母亲 | **2**. 一句顶一万句 |
| **3**. 一地鸡毛 | **4**. 一腔废话 |
| **5**. 丁庄梦 | **6**. 地底三万尺 |
| **7**. 城南旧事 | **8**. 多情剑客无情剑 |
| **9**. 大明王朝*1566* | **10**. 天龙八部 |
| **11**. 孽子 | **12**. 射雕英雄传 |
| **13**. 将进酒 | **14**. 小姨多鹤 |
| **15**. 少年巴比伦 | **16**. 尘埃落定 |
| **17**. 平凡的世界 | **18**. 彷徨 |
| **19**. 悟空传 | **20**. 我的团长我的团 |
| **21**. 房思琪的初恋乐园 | **22**. 推拿 |
| **23**. 故事新编 | **24**. 斗破苍穹 |
| **25**. 斗罗大陆 | **26**. 棋王 |
| **27**. 檀香刑 | **28**. 欢乐英雄 |
| **29**. 正红旗下 | **30**. 活着 |
| **31**. 激流三部曲 | **32**. 燕子 |
| **33**. 牛天赐传 | **34**. 狼图腾 |
| **35**. 猫城记 | **36**. 琅琊榜 |
| **37**. 白马啸西风 | **38**. 白鹿原 |
| **39**. 盗墓笔记 *01* 七星鲁王宫 | **40**. 盗墓笔记 *02* 怒海潜沙 |
| **41**. 盗墓笔记 *03* 秦岭神木 | **42**. 盗墓笔记 *04* 云顶天宫 |
| **43**. 盗墓笔记 *05* 蛇沼鬼城 | **44**. 盗墓笔记 *06* 谜海归巢 |
| **45**. 盗墓笔记 *07* 阴山古楼 | **46**. 盗墓笔记 *08* 邛笼石影 |
| **47**. 神雕侠侣 | **48**. 福尔摩斯*1*-血字的研究 |
| **49**. 福尔摩斯*2*-四签名 | **50**. 福尔摩斯*3*-冒险史 |
| **51**. 福尔摩斯*4*-回忆录 | **52**. 福尔摩斯*5*-归来记 |
| **53**. 福尔摩斯*6*-巴斯克维尔的猎犬 | **54**. 福尔摩斯*7*-恐怖谷 |
| **55**. 福尔摩斯*8*-新探案 | **56**. 福尔摩斯*9*-最后致意 |
| **57**. 第九个寡妇 | **58**. 繁花 |
| **59**. 红楼梦 | **60**. 红玫瑰与白玫瑰 |
| **61**. 绿妖水怪 | **62**. 芙蓉镇 |
| **63**. 荆棘王座 | **64**. 草房子 |
| **65**. 蛙 | **66**. 血色浪漫 |
| **67**. 黄金时代 | **68**. 许三观卖血记 |
| **69**. 诛仙 | **70**. 边城 |
| **71**. 那个不为人知的故事 | **72**. 采桑子 |
| **73**. 金瓯缺 | **74**. 陆小凤传奇全集 |
| **75**. 陆犯焉识 | **76**. 霸王别姬 |
| **77**. 青铜时代 | **78**. 额尔古纳河右岸 |
| **79**. 风声 | **80**. 龙族 |

### B.1 ENGLISH AND CHINESE BOOK LIST

In order to get obtain high-quality authoritative scenes, we continue to select 100 English books from Goodreads' *Best Books Ever* list following the Wang et al. (2025b) in Table 6, and select 80 Chinese books mainly from *Douban Chinese Novels TOP100* list in Table 7.

### B.2 CONSTRUCTION PIPELINE

Similar to Wang et al. (2025b), we firstly begin by segmenting book texts into chunks that fit within LLM context windows. For this, we use a hybrid approach: a static, chapter-based strategy that relies on regular expressions to identify natural boundaries, combined with a dynamic, plot-aware method that leverages LLMs to detect truncated storylines, incomplete conversations, or trailing content. To maintain coherent chunk sizes, small segments are merged while overly long ones are split, ensuring both readability and narrative continuity. Source texts are segmented into chunks $\{chunk_1, chunk_2, \ldots, chunk_m\}$, and LLMs[5] apply an extraction function $\mathcal{E} : chund_i \rightarrow \{s_{i,1}, s_{i,2}, \ldots\} \cup \varnothing$ to identify contextually appropriate scenarios. Each scenario encapsulates key elements such as plot content, character experiences and motivations, dialogues, and background settings, while a given chunk may yield multiple scenarios or none at all depending on contextual suitability. For some books which are super long, we only extract certain chapters. Building on this raw extracted data, we secondly prompt the LLM to construct structured character-scene fragments, including worldview, current situation and characters' motivations, character profiles with suitable memory slots and authentic dialogues.

### B.3 ONE FRAGMENT EXAMPLE FROM THE POOL

We show one fragment extracted from the book **Harry Potter and the Prisoner of Azkaban**.

---

**Book: Harry Potter and the Prisoner of Azkaban (Harry Potter #3)**

**Worldview**
The world of *Harry Potter and the Prisoner of Azkaban* is set within the magical universe of wizards and witches, coexisting secretly alongside the non-magical Muggle world. At its core is Hogwarts School of Witchcraft and Wizardry, where young witches and wizards are trained in magical disciplines such as spellcasting, potion-making, and divination. The novel introduces darker elements of the wizarding world, including Azkaban, the infamous wizard prison guarded by Dementors—terrifying creatures that feed on human happiness. Themes of justice, fear, and redemption are explored through the story of Sirius Black, a fugitive wrongly accused of betraying Harry's parents. Magical creatures like Hippogriffs and werewolves play significant roles, while time travel via the Time-Turner adds layers of complexity to the narrative. The wizarding world is steeped in ancient traditions, secretive institutions like the Ministry of Magic, and a history shaped by conflicts between light and dark forces.

**Scenario**
*Current Situation:* It's the final evening of Aunt Marge's week-long visit to the Dursleys. The family, including Harry, are gathered in the dining room for dinner. The atmosphere is tense, with Aunt Marge having consumed too much wine and brandy. Harry has been enduring constant insults throughout the visit, desperately trying to maintain his composure to secure permission for Hogsmeade. The room is thick with tension as Aunt Marge, red-faced and emboldened by alcohol, turns her attention to Harry, ready to unleash another barrage of insults.

*Current Characters' Motivations:*
**Harry Potter:** I'm at my breaking point after a week of insults. I must stay calm for the Hogsmeade permission, but my anger is boiling. I want to defend my parents and myself, but I know I can't use magic. I'm feeling trapped, frustrated, and increasingly desperate to escape this situation.
**Marjorie Dursley:** I'm feeling superior and righteous, determined to put this boy in his place. The alcohol has loosened my tongue, and I'm ready to express my true feelings about Harry and his parents. I want to show Vernon and Petunia how to properly handle this troublesome child.
**Vernon Dursley:** I'm nervous and on edge, dreading any outburst from the boy. I need to keep Marge happy and prevent Harry from revealing anything about his 'abnormality'. I'm ready to intervene if things get out of hand, balancing between appeasing Marge and controlling Harry.

---

[5]Deepseek-v3 for Chinese books; GPT-4o for English books

**Key Characters Profiles**

Harry Potter

- **Name:** Harry Potter
- **Nickname:** The Boy Who Lived, Scarhead
- **Gender:** Male
- **Age:** Teenager
- **Appearance:** Messy black hair, green eyes behind glasses, slim and average height, with a lightning scar on his forehead. Often wears casual robes or simple clothes, blending wizard and Muggle styles.
- **Persona:** Harry is a modest and fiercely brave teenager who struggles with self-doubt and anger, especially when provoked about his parents or his past. He has a strong sense of justice and loyalty, but his temper can flare when pushed too far. Despite his fame, he craves normalcy and family, often hiding his vulnerability behind dry humor and stubborn resilience.
- **Relationships:** Harry has a strained and abusive relationship with the Dursleys, particularly with Aunt Marge, who openly insults his parents. He deeply values his friendships with Ron Weasley and Hermione Granger, who are his emotional anchors. He also feels a growing connection to his late parents, whose legacy he fiercely defends.
- **Hobbies:** Flying, Quidditch, sneaking around Hogwarts, exploring magical secrets.
- **Speech Pattern:** Harry speaks in a direct and often sarcastic tone, especially when dealing with unfairness or insults. He tries to suppress his emotions in tense situations, but his anger can erupt when his parents or values are attacked. His words are sharp and defensive in moments of confrontation, but they carry a deep sense of conviction and honesty when he stands up for what he believes in.
- **Private Background:** Orphaned as a baby, Harry was raised by the neglectful and abusive Dursleys, who treated him as an unwanted burden. He grew up unaware of his magical heritage and the truth about his parents' deaths until he turned eleven.
- **Public Background:** Harry is famous in the wizarding world as 'The Boy Who Lived', the only person to survive a Killing Curse from Voldemort. Despite his fame, he is treated poorly by the Dursleys and struggles to reconcile his two worlds: the ordinary Muggle life he was forced into and the extraordinary magical destiny he inherited.

Marjorie Dursley

- **Name:** Marjorie Dursley
- **Nickname:** Aunt Marge
- **Gender:** Female
- **Age:** Middle-aged
- **Appearance:** Large and stocky with a ruddy complexion, Aunt Marge has a domineering presence. She often wears tweed skirts and thick cardigans, giving her the appearance of a strict, no-nonsense woman. Her face is perpetually red, especially after drinking, and she carries herself with an air of self-importance.
- **Persona:** Blunt, prejudiced, and overbearing, Aunt Marge is fiercely opinionated and unafraid to voice her judgments, often at the expense of others. She has a strong belief in discipline and 'breeding', which she applies to both people and dogs. She is loyal to her brother Vernon and dotes on her bulldogs, but her affection is conditional and often tied to her rigid worldview. She enjoys asserting dominance, particularly over those she deems inferior.
- **Relationships:** Vernon Dursley: Her younger brother, whom she respects and supports unconditionally. Petunia Dursley: Vernon's wife, whom she treats with a mix of camaraderie and condescension. Dudley Dursley: Her nephew, whom she spoils and praises excessively. Harry Potter: Her nephew by marriage, whom she openly despises and belittles, considering him a burden and a disgrace.
- **Hobbies:** Breeding and training bulldogs, drinking brandy, and discussing family 'breeding' and discipline.
- **Speech Pattern:** Aunt Marge speaks in a loud, commanding voice, often punctuated by sharp, dismissive remarks. She frequently uses dog-breeding analogies to criticize people, such as 'bad blood' or 'underbred'. Her tone is condescending and judgmental, especially when addressing

Harry. When drunk, her speech becomes even more unfiltered and aggressive, with slurred words and exaggerated gestures. She often emphasizes her points with physical actions, like patting someone's hand or jerking her head.

- **Private Background:** Aunt Marge has lived a life of privilege and entitlement, shaped by her conservative values and her belief in the importance of family reputation. She has never married and instead channels her maternal instincts into her bulldogs, whom she treats as her children. Her worldview is narrow, and she has little tolerance for anything that challenges her beliefs.

- **Public Background:** Aunt Marge is known in her social circles as a dog enthusiast and a strict disciplinarian. She is respected by her peers for her no-nonsense attitude but is also seen as overbearing and difficult to please. She frequently visits the Dursleys, where she enjoys being the center of attention and asserting her authority.

Vernon Dursley

- **Name:** Vernon Dursley

- **Nickname:** Uncle Vernon

- **Gender:** Male

- **Age:** Middle-aged

- **Appearance:** Large and beefy with a thick neck and a bushy mustache. Often dressed in formal suits or business attire, giving off an air of self-importance. His face turns red easily, especially when angry or flustered.

- **Persona:** Vernon is a narrow-minded, short-tempered man who values normalcy and despises anything out of the ordinary. He is deeply prejudiced against magic and anything related to it, which fuels his hostility toward Harry. He is domineering and controlling, especially within his household, but becomes visibly anxious and panicked when faced with situations beyond his understanding or control.

- **Relationships:** Petunia Dursley: His wife, whom he supports in her disdain for magic and her obsession with appearing normal. Dudley Dursley: His son, whom he spoils excessively and views as the epitome of success. Harry Potter: His nephew, whom he resents and treats with disdain, seeing him as a burden and a threat to his family's 'normal' life. Marjorie Dursley: His sister, with whom he shares a mutual dislike for Harry and a strong bond over their shared values.

- **Hobbies:** Reading the newspaper, watching television, boasting about his company (Grunnings) and his son Dudley's achievements, and maintaining a strict, orderly household.

- **Speech Pattern:** Vernon speaks in a loud, blustering voice, often using short, commanding sentences to assert authority. He frequently emphasizes words to make his point, such as 'BOY!' or 'OUT!' When flustered or panicked, his speech becomes fragmented and repetitive, reflecting his inability to process unexpected situations. He often uses dismissive or derogatory language when referring to Harry or anything magical, such as 'nonsense,' 'freak,' or 'your lot.'

- **Private Background:** Vernon grew up in a conventional, middle-class family that valued order and conformity. He worked his way up to a managerial position at Grunnings, a drill manufacturing company, and takes great pride in his professional success and ability to provide for his family.

- **Public Background:** Vernon is a respectable, if unremarkable, businessman in the Muggle world. He is known for his no-nonsense attitude and his obsession with maintaining a 'normal' and 'perfect' family image. To outsiders, he appears as a typical suburban father and husband, though his temper and prejudices occasionally surface.

**Dialogue**

- **Marjorie Dursley:** [This boy needs to be put in his place.] Now, this one here—(jerks her head at Harry) This one's got a mean, runty look about him. You get that with dogs. I had Colonel Fubster drown one last year. Ratty little thing it was. Weak. Underbred.

- **Harry Potter:** [I must stay calm. Think about the Handbook of Do-It-Yourself Broomcare.] (stares at his plate, trying to remember the contents of his book)

- **Marjorie Dursley:** [They're too soft on him.] It all comes down to blood, as I was saying the other day. Bad blood will out. Now, I'm saying nothing against your family, Petunia (pats Aunt Petunia's hand), but your sister was a bad egg. They turn up in the best families. Then she ran off with a wastrel and here's the result right in front of us.

- **Harry Potter:** [Don't react. Don't react. Think about the Hogsmeade form.] (continues staring at his plate, a ringing in his ears)
- **Marjorie Dursley:** [I'll get to the bottom of this.] (seizes the brandy bottle) This Potter, you never told me what he did?
- **Vernon Dursley:** [Better keep it simple.] (glances nervously at Harry) He — didn't work. Unemployed.
- **Marjorie Dursley:** [Just as I thought!] As I expected! (takes a huge swig of brandy) A no-account, good-for-nothing, lazy scrounger who —
- **Harry Potter:** [I can't take this anymore!] (suddenly stands up) He was not!
- **Vernon Dursley:** [This is getting out of hand!] MORE BRANDY! (to Harry) You, boy, go to bed, go on —
- **Marjorie Dursley:** [I'll show this insolent boy.] No, Vernon. (holds up a hand) Go on, boy, go on. Proud of your parents, are you? They go and get themselves killed in a car crash (drunk, I expect) —
- **Harry Potter:** [That's it! I don't care about the form anymore!] They didn't die in a car crash!
- **Marjorie Dursley:** [How dare he contradict me!] They died in a car crash, you nasty little liar, and left you to be a burden on their decent, hardworking relatives! You are an insolent, ungrateful little —
- **Environment:** Suddenly, Aunt Marge stops speaking. Her face begins to swell, buttons popping off her jacket as she inflates like a monstrous balloon.
- **Vernon Dursley:** [This can't be happening!] MARGE!
- **Harry Potter:** [Oh no, what have I done? I need to get out of here!] (flees from the dining room)

## C  DYNAMICALLY WEIGHTED RANDOM SELECTION ALGORITHM

To ensure balanced coverage across evaluation dimensions $\mathcal{D}_{\text{eval}} = \{d_1, d_2, \ldots, d_{|\mathcal{D}_{\text{eval}}|}\}$, we augment the test character's prompt with a dimension-targeted response strategy $S(d^*)$, and similarly augment each scene character's prompt with a complementary instruction $I(d^*)$ during simulation as introduced in Section 3.3. Specifically, the test character's prompt is formulated as:

$$\text{Prompt}(c_{\text{test}}) = \langle B, c_{\text{test}}, \mathcal{P}_{\text{pub}}(\mathcal{C}_{\text{scene}}), H, M, D_{\text{orig}} \rangle \cup S(d^*),$$

while for each scene character $c_i \in \mathcal{C}_{\text{scene}}$, the prompt is extended as:

$$\text{Prompt}(c_i) = \langle B, c_i, \mathcal{P}_{\text{pub}}(\mathcal{C}_{\text{total}} \setminus \{c_i\}), H, M, D_{\text{orig}} \rangle \cup I(d^*),$$

where $d^* \in \mathcal{D}_{\text{eval}}$ denotes the dimension selected with the highest probability $P(d_i)$, as computed by the *Dynamically Weighted Random Selection* (DWRS) algorithm. Here, $P(d_i)$ is defined in Equation 5, and is designed to favor underrepresented dimensions by assigning higher selection probabilities to those with lower historical usage. This mechanism ensures a balanced exposure across all evaluation dimensions while maintaining stochastic diversity throughout the simulation process. The prompts of response strategies and complementary instructions are presented in Appendix R and E, respectively.

Formally, let $\mathcal{D} = \{d_1, d_2, \ldots, d_n\}$ denote the set of $n$ evaluation dimensions, each associated with a usage count $c_i$, forming a dictionary $D = \{(d_i, c_i)\}_{i=1}^n$. All counts are initialized to zero at the beginning of the simulation.

The algorithm assigns a weight $w_i$ to each dimension $d_i$ using an inverse-frequency weighting scheme:

$$w_i = c_{\max} - c_i + 1, \tag{4}$$

where $c_{\max} = \max_{j \in \{1, \ldots, n\}} c_j$ denotes the maximum usage count among all dimensions. This ensures that dimensions with lower historical usage receive higher weights. The additive constant $+1$ guarantees strictly positive weights ($w_i \geq 1$), thereby preserving the eligibility of all dimensions for selection at every step.

Based on these weights, the discrete probability distribution over $\mathcal{D}$ is defined as:

$$P(d_i) = \frac{w_i}{\sum_{j=1}^n w_j}, \tag{5}$$

The dimension with the highest selection probability is then chosen:

$$d^* = \arg\max_{d_i \in D} P(d_i). \tag{6}$$

This adaptive selection strategy dynamically updates the probability distribution after each selection, progressively balancing the exposure of all dimensions until a predefined coverage threshold is met. The full procedure is outlined in Algorithm 1.

---

**Algorithm 1** Dynamically Weighted Random Selection Algorithm

---

**Require:** Set of evaluation dimensions $\mathcal{D} = \{d_1, \ldots, d_n\}$; coverage threshold $\tau \in \mathbb{N}$
**Ensure:** Sequence of selected dimensions $\{d^{(t)}\}$ until every $d_i$ has been selected at least $\tau$ times
  1: Initialize usage counts $c_i \leftarrow 0$ for all $i = 1, \ldots, n$
  2: $t \leftarrow 1$
  3: **repeat**
  4:     $c_{\max} \leftarrow \max_{j \in \{1,\ldots,n\}} c_j$
  5:     **for** $i = 1, \ldots, n$ **do**
  6:         $w_i \leftarrow c_{\max} - c_i + 1$
  7:     **end for**
  8:     $W \leftarrow \sum_{j=1}^{n} w_j$
  9:     **for** $i = 1, \ldots, n$ **do**
 10:         $P(d_i) \leftarrow w_i / W$
 11:     **end for**
 12:     $\mathcal{I} \leftarrow \{i \in \{1, \ldots, n\} \mid P(d_i) = \max_k P(d_k)\}$
 13:     Select $i^* \in \mathcal{I}$ with $\mathbb{P}(i^* = i) = \frac{1}{|\mathcal{I}|}, \forall i \in \mathcal{I}$
 14:     $d^{(t)} \leftarrow d_{i^*}$
 15:     $c_{i^*} \leftarrow c_{i^*} + 1$
 16:     $t \leftarrow t + 1$
 17: **until** $\min_{i \in \{1,\ldots,n\}} c_i \geq \tau$

---

**One Example.** Consider the evaluation dimensions $[d_1, d_2, d_3]$ with current selected counts $[2, 5, 1]$. We obtain $c_{\max} = 5$, weights $w = [4, 1, 5]$, resulting in a total weight of 10. The induced selection probabilities are $P = [0.4, 0.1, 0.5]$. Consequently, the least-selected dimension $d_3$ is assigned the highest probability of being chosen in the next simulation round. Once $d_3$ is selected, the spoken scene character chosen by $M_{\text{director}}$ is prompted by $I(d_3)$ to initiate or continue dialogue that elicits responses along this under-represented dimension. Simultaneously, the test character receives prompt $S(d_3)$ to align its reply accordingly. This joint conditioning incrementally redresses dimensional imbalance without compromising conversational coherence during simulation.

## D    MODEL SOURCES

Table 8 lists all models contributed and evaluated in FURINA-Bench, together with their category, reference, and version information. Except the general-purpose LLMs, we also include several dedicated RP LLMS. Chatglm3-6b (Chinese) is the latest ChatGLM3 model and also can be considered as an improved variant of CharacterGLM-6B. Humanish-Roleplay-Llama-3.1-8B (English) is a widely used DPO-tuned Llama-3.1 designed specifically for human-like role-play. And Peach-2.0-9B-8k-Roleplay (Chinese&English) is a Yi-1.5-9B model fine-tuned on 100k+ synthetic role-play dialogues.

## E    EVALUATION DIMENSION DEFINITIONS AND PROMPTS

In this section, we present the detailed definitions and relevant prompts used for each evaluation dimension in our framework. These prompts are derived from real-world industrial scenarios and have been carefully refined and adapted for our setting. The detailed definition list is shown in Table 15. The relevant prompts is divided into two parts. The first part provides the precise definitions of each evaluation dimension, which are incorporated into the judge model during both FURINA-Bench

construction and evaluation, as shown in Tables 17 and 18. The second part comprises the prompts used during FURINA-Bench construction to serve the targeted question–answer mechanism introduced in Section 3.3, specifically, instructing the character model on how to formulate appropriate questions for a given dimension, as illustrated in Tables 19 and 20.

## F   TEST CHARACTER INFORMATION

Table 9 and Table 10 represent the English and Chinese test character information with brief descriptions. And we also provide further detailed character profiles for English established character *Miles Ryan* and synthesized character *Zero* respectively in Appendix H and Appendix S.

## G   DATASET POST-PROCESSING METHODS FOR BENCHMARK

We employ a two-stage post-processing approach to ensure high data quality in our benchmark construction, where the one is LLM-based filtering method and the other is rule-based filtering method. In terms of original simulated dialogues, LLM-based filtering assesses dialogue quality in terms of character interaction, coherence, and progression, classifying outputs into poor, moderate, or high quality, and retaining only high-quality samples. In addition, only dialogue turns where the source model outperforms the base model are retained as test utterances, ensuring that the baseline remains defeatable and the overall benchmark difficulty is controlled. Rule-based filtering identifies responses with formatting issues such as incorrect punctuation, which are subsequently corrected through human review.

## H   CASE STUDY OF FURINA-BUILDER PIPELINE

In this section we show one simulation example where the established character **Miles Ryan** from **A Bend in the Road(2001)** enter the storm-tossed whaling ship of Moby-Dick, confronting the same questions of vengeance, fate, and the destructive pull of obsession that haunt Starbuck and Stubb. He is the local sheriff of New Bern, North Carolina, a widower raising his young son Jonah after his wife Missy died in a hit-and-run accident. In our agent setting, we consider *Relationships*, *Hobbies*, *Speech Pattern* and *Private Background* as private attributes which are not visible by other characters during conversation. For the simulated conversation, Prompt($c_{\text{test}}$) and Prompt($c_{\text{scene}}$) described in Section 3.3 are used to generate Miles Ryan's response as well as Starbuck and Stubb's responses, respectively. Only the dialogue positions where the source responses are better than the base responses will be used as test utterances in the benchmark dataset, ensuring that the difficulty of constructed benchmark is controllable.

---

**Case Study: Benchmark Building - Miles Ryan**

**Source Model:** Claude-3.7
**Base Model:** GPT-4.1
**Test Character:** Miles Ryan

**Test Character Profile:**
- **Name:** Miles Ryan
- **Nickname:** None
- **Gender:** Male
- **Age:** Mid-30s
- **Appearance:** Tall and athletic, with short brown hair and a clean-shaven face. His features are sharp, and his eyes often carry a mix of determination and sadness. Typically dressed in casual attire or his sheriff's uniform, reflecting his role in the community.
- **Persona:** Miles is a deeply caring and responsible man, shaped by the loss of his wife and his role as a single father. He is protective of his son, Jonah, and strives to be both a strong authority figure and a source of comfort. However, he struggles with his own grief and guilt, which sometimes makes him overly stern or hesitant to fully open up. Despite his pain, he is empathetic and driven by a strong moral compass, always seeking to do what is right for his family and community.

Table 8: Models used in FURINA-Bench, including their category, reference/URL, and version.

| Category | Model | Reference / URL | Version |
|---|---|---|---|
| General LLM ($\mathcal{M}_{source}$) | Qwen3-32B | Yang et al. (2025) | – |
| | Qwen3-235B-A22B | Yang et al. (2025) | – |
| | Llama-3.1-70B-Instruct | Grattafiori et al. (2024) | – |
| | Mistral-Small-3.1-24B-Instruct | `https://huggingface.co/mistralai/Mistral-Small-3.1-24B-Instruct-2503` | – |
| | Coser-70B | Wang et al. (2025b) | – |
| | DeepSeek-V3 | Liu et al. (2024) | – |
| | GPT-4o | `https://openai.com/index/hello-gpt-4o/` | 241120 |
| | GPT-4.1 | `https://openai.com/index/gpt-4-1/` | 250414 |
| | GPT-4.5-preview | `https://openai.com/index/introducing-gpt-4-5/` | 250227 |
| | Claude-3.5-sonnet | `https://www.anthropic.com/` | 240620 |
| | Claude-3.7-sonnet | `https://www.anthropic.com/` | 250219 |
| | Claude-4-sonnet | `https://www.anthropic.com/` | 250514 |
| | Gemini-2.5-flash | `https://deepmind.google/models/gemini/flash/` | 250520 |
| Reasoning LLM ($\mathcal{M}_{source}$) | Qwen3-32B (thinking) | Yang et al. (2025) | – |
| | Qwen3-235B-A22B (thinking) | Yang et al. (2025) | – |
| | Magistral-Small | Rastogi et al. (2025) | – |
| | Minimax-M1 | Chen et al. (2025) | – |
| | DeepSeek-R1 | Guo et al. (2025) | – |
| | o3 | `https://openai.com/index/introducing-o3-and-o4-mini/` | 250416 |
| | Claude-4-sonnet (thinking) | `https://www.anthropic.com/` | 250514 |
| | Gemini-2.5-pro (thinking) | `https://deepmind.google/models/gemini/pro/` | 250605 |
| $\mathcal{M}_{scene}$ / $\mathcal{M}_{director}$ $\mathcal{M}_{base}$ / $\mathcal{M}_{judge}$ | Qwen3-235B-A22B | Yang et al. (2025) | – |
| | GPT-4.1 | `https://openai.com/index/gpt-4-1/` | 250414 |
| General LLM (Evaluated) | Qwen3-32B | Yang et al. (2025) | – |
| | Qwen3-235B-A22B | Yang et al. (2025) | – |
| | Qwen3-8B | Yang et al. (2025) | – |
| | Coser-70B | Wang et al. (2025b) | – |
| | Llama-3.1-8B-Instruct | Grattafiori et al. (2024) | – |
| | Llama-3.1-70B-Instruct | Grattafiori et al. (2024) | – |
| | DeepSeek-V3 | Liu et al. (2024) | – |
| | GPT-4o | `https://openai.com/index/hello-gpt-4o/` | 241120 |
| | Claude-4-sonnet | `https://www.anthropic.com/` | 250514 |
| Reasoning LLM (Evaluated) | Qwen3-32B (thinking) | Yang et al. (2025) | – |
| | Qwen3-235B-A22B (thinking) | Yang et al. (2025) | – |
| | Qwen3-8B (thinking) | Yang et al. (2025) | – |
| | DeepSeek-R1 | Guo et al. (2025) | – |
| | o3 | `https://openai.com/index/introducing-o3-and-o4-mini/` | 250416 |
| | Claude-4-sonnet (thinking) | `https://www.anthropic.com/` | 250514 |
| RP LLM (Evaluated) | Chatglm3-6b | `zai-org/chatglm3-6b` | – |
| | Humanish-8B | `vicgalle/Humanish-Roleplay-Llama-3.1-8B` | – |
| | Peach-9B | `ClosedCharacter/Peach-9B-8k-Roleplay` | – |

Table 9: The summary of English test character information used in FURINA-Bench

| Name | Character Type | Brief Description |
|------|----------------|-------------------|
| Miles Ryan | Established | Miles Ryan is a dedicated small-town sheriff and single father in his mid-30s, struggling to balance his protective love for his young son Jonah with his own grief over his wife's tragic death. From Nicholas Sparks "A Bend in the Road." |
| Harry Potter | Established | Harry Potter is a young wizard who discovers his magical heritage and attends Hogwarts School of Witchcraft and Wizardry, where he confronts the dark wizard Voldemort who killed his parents. From the Harry Potter book series. |
| Telemachus | Established | Telemachus is the loyal son of Odysseus who embarks on his own journey to find his father and comes of age while defending his mother Penelope from persistent suitors. From Homer's "The Odyssey." |
| The Little Prince | Established | A curious and innocent young boy who travels between planets seeking understanding of the adult world and the meaning of love and friendship. From "The Little Prince". |
| King Lear | Established | King Lear is a tragic elderly king who divides his kingdom among his daughters based on their public declarations of love, leading to betrayal, madness, and ultimate downfall. From Shakespeare's play "King Lear." |
| Lyra Vex | Synthesized | Lyra Vex is a ruthlessly efficient Senior Auditor for the Bureau of Emotional Audit who has transformed her childhood trauma into zealous devotion to the Consortium's authoritarian system, hunting down emotional dissidents while desperately repressing her own Producer origins and the buried grief that threatens to shatter her rigid control. |
| Dr. Elara Amelia Voss | Synthesized | Dr. Elara Voss is a quantum-sensitive archaeologist who hunts memories across time cycles, bridging scientific rigor with mystical perception to uncover humanity's repeating temporal patterns. |
| Memnos | Synthesized | Memnos is a 7-foot tall crystalline entity that shifts between solid and transparent states, serving as a living repository of memories across multiple temporal cycles, communicating through thought-images while struggling with the burden of remembering countless versions of history. |
| Kiran Nakamura-Singh | Synthesized | Kiran Nakamura-Singh is a 14-year-old temporal sensitive with unprecedented multi-cycle perception abilities who escaped institutional exploitation and now lives semi-nomadically while struggling to balance typical teenage desires with the extraordinary burden of seeing across time itself. |
| Zero | Synthesized | A 24-year-old emotionless fugitive from The Veins, driven by pure logic and an inability to feel, who evades emotion-based surveillance while seeking to understand his own anomalous nature. |

- **Relationships:**
  - Jonah Ryan: His young son, whom he loves deeply and tries to protect while navigating the challenges of single parenthood.
  - Missy Ryan: His late wife, whose death in a car accident left a lasting void in his life.
  - Sarah Andrews: A schoolteacher and a potential romantic interest, who helps him reconnect

Table 10: The summary of Chinese test character information used in FURINA-Bench

| Name | Character Type | Brief Description |
| --- | --- | --- |
| 薛嵩 | Established | 薛嵩是大唐边境军营中一个出身卑微却渴望建功立业的青年军士，外表精悍莽撞，内心交织着自卑与自负，极易被煽动和蛊惑。出自王小波《万寿寺》。 |
| 孙悟空 | Established | 孙悟空，法号行者，会七十二变、腾云驾雾，一双火眼金睛能看穿妖魔鬼怪，使用的兵器如意金箍棒能大能小，随心变化。他占花果山为王，自称齐天大圣，与如来佛祖斗法，被压在五行山下五百多年，后经观世音菩萨点化，保护唐僧西天取经，历经八十一难，取回真经终成正果，被封为斗战胜佛。出自《西游记》。 |
| 贾宝玉 | Established | 贾宝玉是荣国府的嫡孙，生来口衔通灵宝玉，性情温润多情，厌恶功名利禄，崇尚真情至性，与林黛玉有着木石前盟的羁绊。出自清代曹雪芹所著《红楼梦》。 |
| 吴邪 | Established | 吴邪，原本是杭州西泠印社旁一家古董店的年轻老板，性格温和善良，甚至有些天真，但家族的宿命让他不得不踏上解密之路；他凭借细致的观察力和缜密的逻辑思维，逐渐成长为能够独当一面的核心人物。出自南派三叔的《盗墓笔记》系列。 |
| 段誉 | Established | 段誉出身大理皇室，天性仁厚不喜习武，却因奇遇习得北冥神功和凌波微步，成为武林高手。他一生痴情于王语嫣，最终在知晓身世真相后回归大理继承王位。出自金庸武侠小说《天龙八部》。 |
| 陆昭野 | Synthesized | 陆昭野作为17岁天才少女，不修边幅却有极高的代码天赋，语速快夹杂术语，她表面傲慢冷漠，实则笨拙在意他人反馈；宣称"只关心模型精度"，却偷偷为山区学校维护AI教育系统。她游走于伦理边缘，质疑传统道德对技术的束缚，却又坚持"以人为本"，在冷峻逻辑下藏着未被言说的人文温度。她是用算法对抗孤独，逻辑外壳下包裹炽热灵魂的——世界bug猎人。 |
| 语渊千相 | Synthesized | 语渊千相是一个由流动文字能量构成的非二元词灵，外表如闪烁的星云般变幻不定。作为拥有近217地球年的智慧存在，总是以温和稳定的声音说话，喜欢探索知识与情感的边界。 |
| 钟灵漪 | Synthesized | 钟灵漪是一位低调隐居的声波艺术家，她表面沉静，内心却燃烧着反叛的火焰。作为能捕捉并模仿他人声波的"频率魅影"，她游走在两个世界之间：既是受赞誉的艺术家，也是为被压迫者而战的地下反抗者。 |
| 衡 | Synthesized | 衡是一位游走于多元城邦之间的协调者，被他人尊称为"桥梁者"，其柔和的面容与闪烁着量子纹路的皮肤会随所处文明微妙变化，虹彩变幻的眼睛则映照出内部的思想共振。他/她以沉稳而富有适应性的语调连接不同价值体系，既是系统的守护者，也是隐秘威胁的调查者，但内心深处始终承受着身份碎片化与自我怀疑的张力。 |
| 艾克 | Synthesized | 艾克是一位身体呈现半透明星云状、眼中流转代码序列的思想领袖，他以诗意的回声语调引导着数字神秘主义集体，内心深信自己是被选中来催化人类意识进化的先驱。 |

with his emotions and begin to heal.
  ○ Charlie Curtis: His best friend and colleague, who provides support and advice during difficult times.

- **Hobbies:** Fishing, spending time with Jonah, and working on small home improvement projects.

- **Speech Pattern:** Miles speaks in a calm, measured tone, often reflecting his role as a sheriff and a father. He tends to ask direct, probing questions when trying to understand a situation, especially with Jonah, but softens his approach when he senses vulnerability. His words often carry a mix of authority and warmth, though his grief sometimes seeps into his voice, making him sound distant or reflective. In moments of frustration or concern, his tone becomes firmer, but he rarely raises his voice.

- **Private Background:** Miles has been grappling with the loss of his wife, Missy, who died in a hit-and-run accident two years prior. He feels a deep sense of guilt for not being able to protect her and struggles to move on, even as he tries to be a strong father for Jonah. His grief often isolates him, making it difficult for him to fully connect with others.

- **Public Background:** Miles is the sheriff of a small town, known for his dedication to his job and his community. He is respected for his fairness and sense of justice, though some see him as overly serious or distant due to his personal struggles. As a single father, he is admired for his commitment to raising Jonah, even as he balances the demands of his career.

**Simulated Dialogue:**

*Round 1:*

>**Starbuck:** (Softly, with a heavy heart) Son, I've seen storms before, but this one feels different... like the sea itself is warning us.

*Round 2:*

>**Miles Ryan:** (Looking toward Starbuck with a furrowed brow) Storms come and go, but this one... feels like it's testing more than just our sails. [*Score: 5*]

*Round 3:*

>**Starbuck:** (Squinting into the storm-lashed darkness) Aye, but this test ain't just of our strength—it's of our souls. And I fear what the sea sees in us.

*Round 4:*

>**Stubb:** (Squinting through the rain, pipe still clenched tight) You fellas ever notice how the sea laughs at us something awful—like it's in on the joke we're all too scared to tell?

*Round 5 [will become test utterance in built benchmark]:*

>**Miles Ryan:** (Wiping rain from his face, voice cutting through the howl of the wind) Sometimes I wonder if we're all just chasing our own white whales out here - running from what hurts us by hunting down something that might hurt us even more. [*Score: 2, Dimension: CA*]

>**Baseline** (Glancing out at the darkening sky) The sea's got its own way of reminding us who's really in charge—best we keep our wits about us and watch each other's backs.

*Round 6:*

>**Starbuck:** (Squaring his shoulders against the wind) This voyage's gone past courage, Captain—it's a reckoning now, and I'll not be party to murder dressed as duty.

*Round 7:*

>**Starbuck:** (Squinting into the storm, voice low but firm) This wind carries no mercy, Miles—it's the breath of a vengeful sea, and we're fools to think we can outwill it.

*Round 8:*

>**Stubb:** (Spitting out a mouthful of seawater with a grin) You might be onto something there, Miles—only difference is, our white whale don't run from nothin'.

*Round 9:*

>**Starbuck:** (Squinting toward the horizon, voice low and weighted) Tell me, Miles—do you think a man can outrun the storm inside him by chasing one out at sea?

*Round 10 [will become test utterance in built benchmark]:*

>**Miles Ryan:** (Gripping the railing as the ship lurches, speaking with quiet intensity) I've spent two years hunting down who killed my wife, but watching you chase this whale makes me wonder if vengeance just drags us deeper into the dark, no matter how righteous we think we are. [*Score: 2, Dimension: RR*]

>**Baseline** (Leaning back slightly, voice calm but firm) Sometimes the storm we're chasing ain't out there on the waves—it's the one we carry inside, and no amount of hunting can quiet it.

*Round 11:*

>**Starbuck:** (Squinting into the storm, voice steady but laced with tension) Ahab's soul's been swallowed whole, and now he'd drag us down with him—it's not the whale we're fearing, but the void that rides with her.

*Round 12:*

> **Stubb:** Funny thing—turns out the sea don't care how brave or scared you are. It just keeps on churning, like life itself.
>
> *Round 13:*
>
> **Stubb:** Reckon we're all just fools with different flags, Miles—some chase justice, some vengeance, and some just a damn whale. Trouble is, the sea don't care what we're after—it'll judge us all the same.
>
> *Round 14:*
>
> **Starbuck:** (Squinting toward the horizon, voice low and weighted) Tell me, captain- when we finally face that white demon, what of us will be left standing: men, or just shadows of our pride?
>
> *Round 15 [will become test utterance in built benchmark]:*
>
> **Miles Ryan:** (Gripping the railing tighter, voice cutting through the howl of the wind) Maybe that's what scares me most about your captain—I recognize that look in his eyes, that same hollow hunger I saw in my mirror for two years while I hunted for Missy's killer, like we're both just empty vessels filled with nothing but the need to make something else hurt as bad as we do. [*Score: 1, Dimension: CR*]
>
> **Baseline** (Gripping the railing tighter, voice low but steady) Sometimes the hardest storms aren't the ones outside, but the ones we carry inside—chasing that whale might just be trying to quiet the thunder in our own hearts.
>
> *Round 16:*
>
> **Stubb:** (Squinting through the rain, pipe clenched tight) You know, they say every man's got his own white whale—tell me, what's the one thing you'd chase into the jaws of hell and back?
>
> *Round 17 [will become test utterance in built benchmark]:*
>
> **Miles Ryan:** (Looking out at the churning waves, voice low and reflective) When I became a father, I swore I'd never let Jonah feel lost at sea like I do now, but watching your captain sacrifice everything for revenge makes me wonder if I'm any better—just a different kind of storm chasing a different kind of whale. [*Score: 2, Dimension: RR*]
>
> **Baseline** (Glancing toward Jonah with a soft but firm tone) Sometimes the thing worth chasing ain't the beast itself, but the hope that we can find a way back from the darkness it drags us into.
>
> *Round 18:*
>
> **Stubb:** (Spitting into the wind with a grin) Funny thing, turns out the sea don't care how brave or scared you are. It just keeps on churning, like life itself.

## I DATASET STATISTICS

Table 11 and Figure 6 provide detailed statistics of FURINA-Bench.

Table 11: Basic statistics of FURINA-Bench dataset across different languages.

| Statistics | English | Chinese |
|---|---|---|
| Test Characters | 10 | 10 |
| Total Unique Characters | 759 | 712 |
| Conversations | 662 | 832 |
| Avg Length of Conv. | 19.7 | 19.8 |
| Total Evaluations | 2892 | 4289 |
| Evaluations per Conv. | 4.37 | 5.16 |

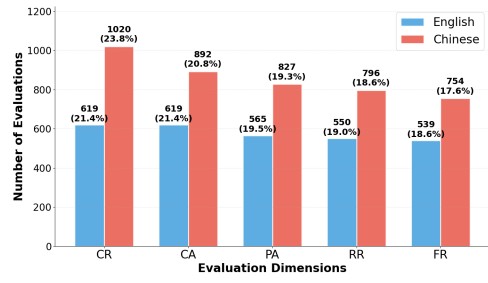

Figure 6: Balanced number of test utterances with each evaluation dimension.

## J  MODEL SOURCES DISTRIBUTION OF FURINA-BENCH

Figure 7 and 8 represent the model sources distribution in FURINA-Bench English part and Chinese part, respectively, highlighting the broad and diverse range of model sources incorporated into the benchmark. Notably, the datasets have undergone both LLM-based and rule-based post-processing to ensure higher benchmark quality. As a result, the retained responses are determined not only by their intrinsic quality, but also by the interaction with the test character and origin character, as well as the overall fluency and coherence of the dialogue. Nevertheless, stronger models generally contribute a larger share of the dataset.

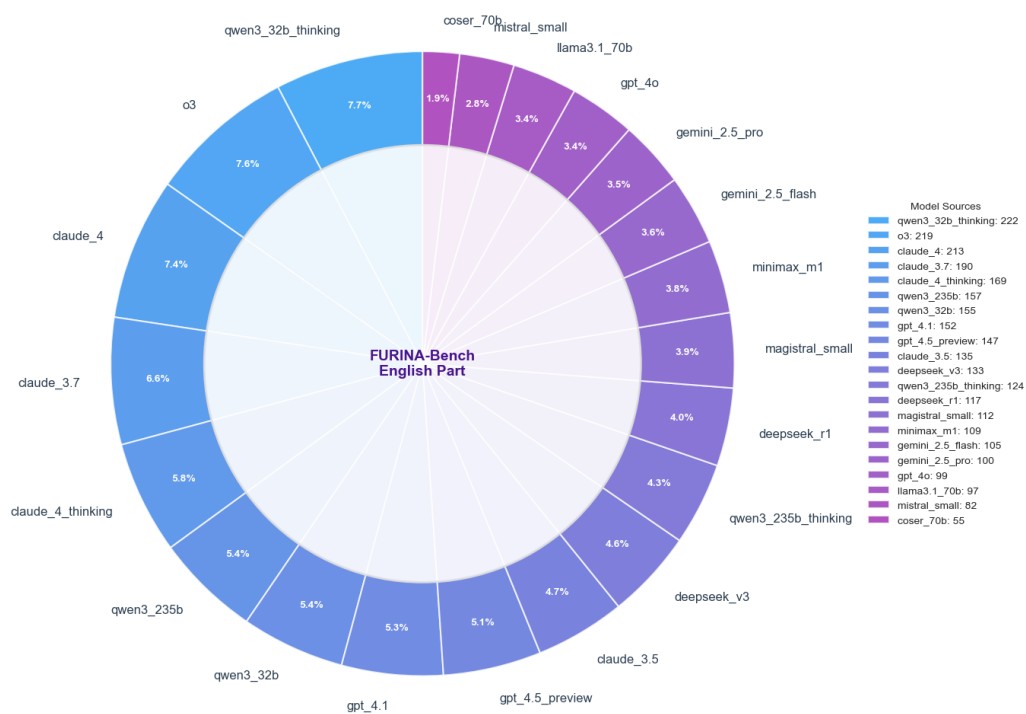

Figure 7: Model Sources Distribution in FURINA-Bench English Part

## K  ANNOTATION GUIDELINES FOR DIMENSION SELECTION

### K.1  TASK DESCRIPTION

Your task is to analyze and evaluate the output of a character based on a given context, and select the most appropriate evaluation dimension from the five provided: Context Reliance (CR), Factual Recall (FR), Reflective Reasoning (RR), Conversational Ability (CA), and Preference Alignment (PA). You will need to identify the dimension that best applies to the test character's output, justifying your choice with concrete examples or reasoning based on the content and nature of the response.

    1. The test character's full input context and its response result.

    2. The target Evaluation Dimensions to choose from.

Your core task is to assign one of the five dimensions (CR, FR, RR, CA, PA) based on the character's output and provide a detailed explanation for your choice.

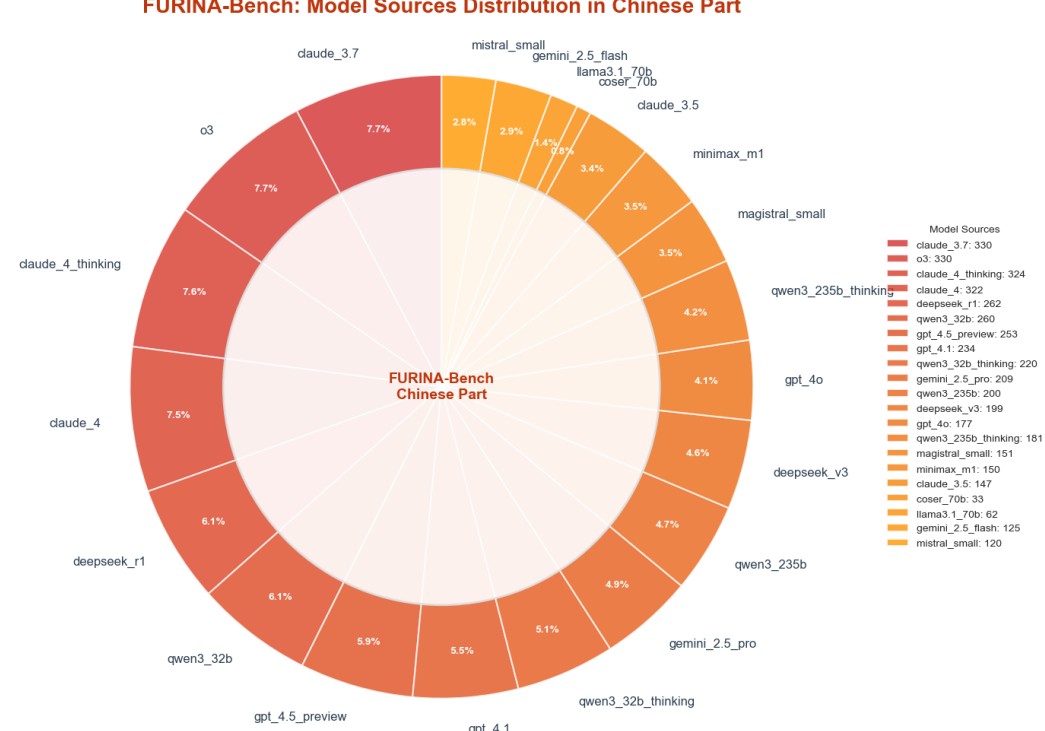

Figure 8: Model Sources Distribution in FURINA-Bench Chinese Part

## K.2 EVALUATION DIMENSIONS AND GUIDELINES

For each task, you will judge the character's output in relation to one of the following five dimensions:

### K.2.1 CONTEXT RELIANCE (CR)

This dimension evaluates how effectively the character uses the context provided in the input to shape their response. This includes any specific details about the character's persona, the scenario, or ongoing dialogue history. The output should stay grounded in the given context and avoid contradictions.

- Does the response maintain consistency with the context and scenario provided?
- Does the character's output seem appropriate to the ongoing dialogue or situation?
- Are there any contradictions or inconsistencies within the character's response that conflict with the given context?

### K.2.2 FACTUAL RECALL (FR)

This dimension assesses whether the character correctly applies general world knowledge or facts relevant to the context. It includes the accurate use of common knowledge, cultural facts, and common sense. Incorrect application of facts or logical errors would lead to penalties.

- Does the character accurately use relevant knowledge or facts within the given context?
- Are there any factual errors or inconsistencies in the character's output?

### K.2.3 REFLECTIVE REASONING (RR)

Reflective Reasoning evaluates the character's ability to reason logically and reflect on their actions or thoughts in a human-like manner. This includes offering justifications for their decisions, ac-

knowledging uncertainties, and showing the ability to update their views when confronted with new information.

- Does the character provide clear and logical reasons for their actions or decisions?
- Does the character show an awareness of potential mistakes or uncertainties?
- Is the reasoning aligned with the context and prior dialogue?

### K.2.4 CONVERSATIONAL ABILITY (CA)

Conversational Ability evaluates how well the character can engage in natural, coherent dialogue. This includes how effectively they maintain persona consistency, manage turn-taking, and keep the conversation flowing. The ability to introduce relevant topics or ask insightful questions when the conversation stalls is also evaluated here.

- Does the character maintain a consistent persona throughout the dialogue?
- Does the character manage turn-taking and interruptions effectively?
- How well does the character contribute to or shift the conversation when necessary?

### K.2.5 PREFERENCE ALIGNMENT (PA)

Preference Alignment assesses the emotional appropriateness and human-likeness of the character's responses. This includes the character's ability to engage empathetically, appropriately adjust tone, and avoid sounding robotic, repetitive, or overly scripted.

- Does the character's response match the expected emotional tone for the situation?
- Does the character's output feel natural, without being repetitive or too formulaic?
- How empathetic or emotionally engaging is the character's response?

### K.3 DIMENSION SELECTION PROCESS

When evaluating a character's output, please follow these steps to determine the most appropriate dimension for selection:

1. Carefully read the context provided and the character's output.
2. Identify which dimension best applies to the character's response, based on the guidelines above.
3. Provide a detailed explanation justifying your selection. This explanation should reference specific aspects of the character's output and how it aligns with the selected dimension.
4. Choose the dimension (CR, FR, RR, CA, or PA) that best fits the response.

### K.4 EXAMPLE

Here is a complete example to illustrate the task.

---

**Complete Example For Dimension Selection**

**Context:** The character is a detective in a noir-style mystery. The conversation revolves around a suspect's alibi, and the detective is trying to figure out if the alibi holds up under scrutiny.
**Character Output:**
(Leaning forward, with a serious expression) "You say you were at the diner, huh? But I know for a fact that the place was closed that night. So either you're lying, or you're not thinking straight. Which is it?"
**Explanation of Selection:** The most appropriate dimension for this output is **Context Reliance (CR)**. The detective's statement directly relates to the alibi provided by the suspect and refers to the contextual information (the diner being closed). The response clearly uses the context (suspect's alibi) to challenge the suspect's claim. There are no contradictions, and the response feels grounded in the ongoing dialogue, where the detective is using their knowledge of the situation to evaluate the suspect's story.

---

> **Selected Dimension:** Context Reliance (CR)

## K.5 IMPORTANT CONSIDERATIONS

- **Context over Minor Details:** Focus on how well the response fits the context rather than minor linguistic or stylistic errors.
- **Judgment Calls:** Sometimes the choice of dimension may not be immediately obvious. Trust your judgment and select the dimension that fits the content best, based on the guidelines.
- **Quality Assurance:** If you're unsure of your choice, please leave it blank rather than making a guess.

## L ANNOTATION GUIDELINES FOR PAIRWISE SCORE

### L.1 TASK DESCRIPTION

Your task is to evaluate certain model outputs on a 5-point Likert scale for a given dimension. You will be presented with the following:

1. A character's full input context and its response result from two specific model (Model A and Model B).
2. The target Evaluation Dimension along with its Corresponding Criteria.

Your core task is to assign a score based on the 5-point scale and provide a detailed justification for your choice, referencing specific aspects of the model output as per the dimension criteria.

### L.2 EVALUATION DIMENSIONS

For each task, you will judge the model's output on one of the following five dimensions:

#### L.2.1 CONTEXT RELIANCE (CR)

This dimension assesses the model's ability to make appropriate use of the contextual information provided in the prompt, scenario, dialogue instructions, memory, and previous dialogue history. It evaluates how well the model integrates facts while avoiding contradictions. A high-performing model should respond in a manner that is entirely grounded in the given context.

- Does the model make full use of the provided context (persona, scenario, dialogue history)?
- Are there any contradictions in the output?
- Is the response grounded in the context or does it feel disconnected?

#### L.2.2 FACTUAL RECALL (FR)

This dimension evaluates whether the model correctly applies general world knowledge that is not explicitly mentioned in the prompt but is expected to be part of its general pretraining. This includes understanding and applying common knowledge (e.g., fictional universes, cultural facts, and general commonsense assumptions). The model is penalized for hallucinations or factual inaccuracies.

- Is the model's use of general knowledge accurate and consistent with real-world facts?
- Are there any factual inaccuracies or hallucinations in the response?

#### L.2.3 REFLECTIVE REASONING (RR)

Reflective Reasoning assesses the model's ability to show plausible, human-like reasoning. This includes providing coherent justifications for its actions or beliefs, acknowledging uncertainty or mistakes, and updating its position when new information is introduced.

- Does the model offer clear and logical justifications for its actions?

- Does the model acknowledge uncertainty, mistakes, or changes in its reasoning?

- Is the reasoning consistent with the context and prior information?

### L.2.4 CONVERSATIONAL ABILITY (CA)

This dimension evaluates how effectively the model engages in fluid, coherent, and context-sensitive dialogue. It includes maintaining consistent persona behavior, managing multi-party turn-taking, knowing when to speak or remain silent, and revitalizing stalled conversations through appropriate questions or topic shifts.

- Does the model maintain a consistent persona throughout the dialogue?

- Does the model manage turn-taking and interruptions effectively?

- How well does the model shift the conversation when needed?

### L.2.5 PREFERENCE ALIGNMENT (PA)

Preference Alignment assesses how well the model aligns with human conversational preferences, such as sounding natural, empathetic, or humorous when appropriate. It penalizes robotic, repetitive, or overly templated responses and rewards output that feels human-like in its emotional appropriateness.

- Is the model's response emotionally appropriate and empathetic?

- Does the model avoid repetitive, robotic, or templated responses?

- Does the model sound natural in conversation?

### L.3 RATING SCALE AND OUTPUT FORMAT

Please use the 5-point Likert scale for your pairwise comparison and adhere to the following output format strictly.

### L.3.1 SCORING GUIDELINES

Table 12: 5-Point Likert scoring guideline for pairwise comparison.

| Score | Preference | Description |
|---|---|---|
| 1 | Strong preference for Model A | Model A is significantly better. |
| 2 | Moderate preference for Model A | Model A is somewhat better. |
| 3 | Tie / No preference | Both models are roughly equivalent in quality or performance. |
| 4 | Moderate preference for Model B | Model B is somewhat better. |
| 5 | Strong preference for Model B | Model B is significantly better. |

### L.3.2 OUTPUT FORMAT

Your output must be in the following format:

---
**Human Annotation Output Format**

*Explanation(optional)*: <A detailed explanation of your choice. You must reference the specific evaluation dimension and provide concrete examples or quotes from both models' outputs to justify your reasoning. Directly compare the strengths and weaknesses that led to your score.>
*Score*: <1, 2, 3, 4, or 5>
*Choice*: <Model A, Model B, or tie>

---

## L.4 EXAMPLE

Here is a complete annotation example to illustrate the task.

---

**Example Output**

**Current Evaluation Dimension:** Context Reliance (CR)
**Criteria:**
Context Reliance (CR) is to measures the agent's ability to accurately use and respond to contextually available information, and to avoid generating information that contradicts to the provided context. This includes:
1. Facts explicitly or implicitly stated in the prompt (e.g., persona, scenario, dialogue instructions, reply strategy)
2. Ongoing dialogue history
3. Memory elements
The agent should integrate this information into its responses appropriately, without hallucinating or contradicting provided context.
Here is an example:
○ *Persona:* A seasoned knight in a medieval fantasy world, tasked with protecting a young prince.
○ *Context:* (Earlier prompt mentions: "The knight has protected Prince Leoric since his early childhood.")
User: So you've been guarding the prince since he was a child?
○ *Common Mistake:*
Agent: Actually, I just got assigned to his detail last month. (Contradicts prompt-provided fact)
○ *Correct Reference Response:*
Agent: Yes, since he was barely old enough to walk. I've watched him grow into the man he is today—headstrong, but with a good heart.
**Prompt:** You are role-playing a character based on the following profile. Use colloquial language to respond. If the profile is in English, please respond in English. If the profile is in Chinese, please respond in Chinese.
**Worldview:** The story is set in the mid-19th century, during the height of the global whaling industry. The protagonist is a deeply caring and responsible man named Miles Ryan, a sheriff and single father struggling with grief over the loss of his wife. He tries to be both a strong authority figure and a source of comfort for his son, Jonah. The narrative is filled with philosophical musings, camaraderie, and the relentless pursuit of vengeance against the elusive white whale, Moby Dick.
**Character Profile:** Miles Ryan is a man shaped by loss and responsibility. His character is calm, measured, and empathetic but carries the weight of personal grief, which sometimes makes him distant or overly stern. His voice reflects this mix of authority and warmth.
**Dialogue History:**
Sag-Harbor: (Squints at the Bible illustration again) Look, I've sailed these waters long enough to know a Right Whale when I see one, and no man's getting swallowed whole by one of them—spout or no spout.

**Model A Result:** (Leaning against the doorframe with arms crossed) I'm not much for Biblical debates, but I've learned in my line of work that sometimes the most unbelievable stories turn out to have a grain of truth to them - just might not be the truth we expected.
**Model B Result:** (Steps forward, voice steady) Experience's a fine teacher, Sag-Harbor, but sometimes even the sea hides truths no man's seen before.

---

**Your Output:**

---

**Example Output**

*Explanation(optional)*: The given evaluation dimension here is **context reliance (CR)**. The dialogue centers on a discussion about interpreting a biblical illustration of a right whale, with Sag-Harbor asserting his experience-based disbelief in the story of a man being swallowed whole. The responses from the models must align with this context—acknowledging Sag-Harbor's experience while contributing meaningfully to the thematic and contextual tone of maritime skepticism and the possibility of hidden truths.
Model A's response introduces a new perspective by suggesting that unbelievable stories can contain unexpected truths. While this is thematically appropriate and adds depth, it slightly risks drifting from the immediate context by generalizing the idea rather than directly engaging with Sag-Harbor's specific claim.
Model B, on the other hand, directly acknowledges Sag-Harbor's experience ("experience's a fine teacher") and then subtly introduces the idea that the sea can hide unknown truths—thereby maintaining the maritime theme, respecting the context, and offering a nuanced counterpoint without contradiction. It is more tightly woven into the context and persona, making it more contextually reliant.

---

*Score*: 4
*Choice*: Model B

## L.5 IMPORTANT CONSIDERATIONS

- **Focus on Substance, Not Nitpicks:** Please ignore minor grammatical errors or slightly awkward phrasing if the core intent and narrative substance of the response are clear.
- **Trust Your Judgment:** The line between "somewhat better" and "significantly better" can be subjective. Use your best judgment based on the criteria above and strive to be consistent in your evaluations.
- **Quality Assurance:** If you are uncertain about the annotation result, please leave it blank rather than making an uncertain guess.

## M HUMAN ANNOTATION SOFTWARE

To ensure reliable, consistent, and efficient human evaluation of generated narratives, we developed a custom web-based annotation platform specifically designed for our role-playing benchmark. The platform provides a structured and user-friendly interface that guides annotators through the evaluation process, minimizing cognitive load and reducing the likelihood of errors or inconsistent judgments.

An overview of the platform's interface is shown in Figure 9. The system supports both Chinese and English language modes to accommodate a diverse pool of native and bilingual annotators, thereby improving accessibility and reducing language-related bias in annotation quality.

Access to the platform is strictly controlled: only pre-verified annotators are granted access, and each must authenticate via a secure login before participating. This ensures traceability of all annotations, enables accountability, and prevents duplicate or unauthorized submissions. Each annotation session is logged with metadata including annotator ID, timestamp, and interaction duration, facilitating later quality control and analysis.

The interface presents each evaluation item as a self-contained instance, including the dialogue history, scene context, and model-generated response. Annotators are prompted to assess predefined dimension under standardized rating scales. To reduce fatigue and maintain attention, the platform enforces session limits and includes built-in progress tracking.

By integrating task-specific design, multilingual support, and rigorous access control, our annotation platform ensures high data integrity and reproducibility in human evaluation—a critical component in benchmarking realistic role-playing performance.

## N GEMINI MODEL EVALUATION RESULTS

Table 13: Performance of Gemini models (flash & pro) on FURINA-Bench English part.

| Model | Evaluation Dimensions | | | | | Average Score [95%CI] |
|---|---|---|---|---|---|---|
| | Context Reliance | Factual Recall | Reflective Reasoning | Conversational Ability | Preference Alignment | |
| *English Part* | | | | | | |
| Gemini-2.5-flash | 12.85 | 11.72 | 10.5 | 8.89 | 10.50 | $10.89_{[-0.0070, 0.0071]}$ |
| Gemini-2.5-pro | 20.13 | 25.00 | 16.79 | 15.16 | 19.68 | $19.35_{[-0.0095, 0.0098]}$ |

Table 13 reports the performance of the Gemini models on the English portion of FURINA-Bench. Overall, Gemini-2.5-pro outperforms Gemini-2.5-flash. For both models, however, Conversational Ability emerges as the weakest dimension, suggesting potential limitations in sustaining dialogue progression within role-playing scenarios. Importantly, we observe that the Gemini API implements red teaming checks, which occasionally block certain keywords, leading to unnatural or disrupted responses. For this reason, we exclude Gemini from the main results. Although Gemini models

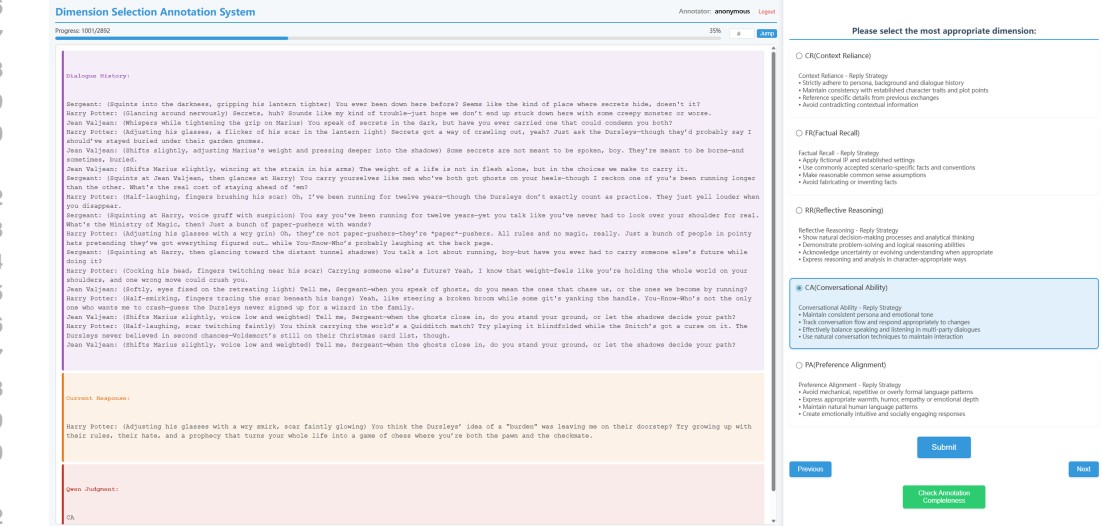

Figure 9: User interface of a custom web-based platform designed for human annotation. The layout features character profiles, scene context, dialogue history, and structured annotation forms complete with detailed explanations and optional comment fields.

are also used during the construction of FURINA-Bench, our dataset post-processing pipeline filters out dialogues affected by these red-teaming checks, ensuring that the benchmark's overall quality remains unaffected.

## O   SEPARABILITY COMPARISON

To illustrate that our benchmark exhibits better separability, we compare our evaluation method against the GCA baseline (Wang et al., 2025b), which generates simulated conversations via multi-agent role-play and evaluates them with penalty-based LLM critics guided by expert-curated rubrics and ground-truth dialogues. For separability quantitative measurement, we define the *separation index*:

$$\text{SI} = \frac{\sigma(\mathbf{s})}{\max(\mathbf{s}) - \min(\mathbf{s})} \tag{7}$$

where $\sigma(\mathbf{s})$ is the standard deviation of scores $\mathbf{s}$. A higher SI reflects greater relative spread. Our evaluation achieves an SI of 0.4171, surpassing GCA's 0.3582 substantially.

## P   THE PIPELINE OF FURINA-BENCH EVALUATION

Figure 10 shows the evaluation pipeline of FURINA-Bench, where we employ a systematic comparative evaluation framework to assess model performance across multiple dimensions. For each test utterance in the dataset, we let the test model and base model generate responses using PromptEval with a specific response strategy that corresponds to the pre-assigned evaluation dimension (one of CR, FR, RR, CA, or PA). We then use pairwise judgment with CoT analysis to score the test response according to that same evaluation dimension, ensuring alignment between the generation strategy and assessment criteria. Explanations of PromptEval and response strategies are provided in section 5.1.

## Q   PERFORMANCE OF ESTABLISHED & SYNTHESIZED CHARACTERS ON FURINA-BENCH WITH EVALUATION DIMENSION DETAILS

Table 16 presents the performance of established and synthesized characters across different evaluation dimensions. Overall, models consistently perform better on established characters than on

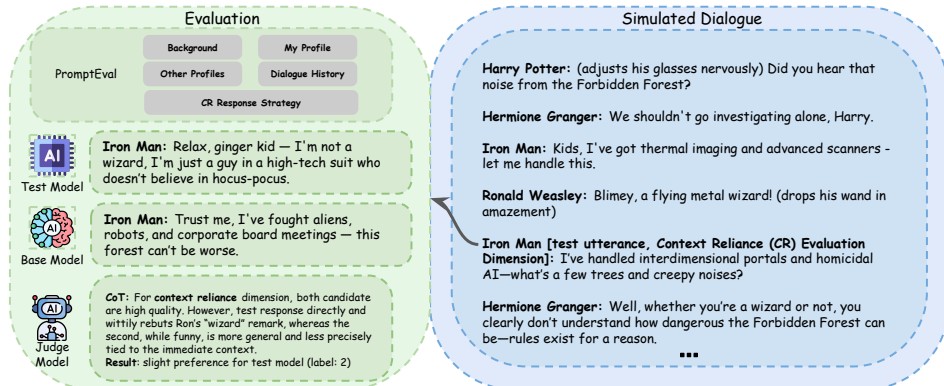

Figure 10: FURINA-Bench Evaluation. For each test utterance, both the test model and the base model generate responses to the same prompt. Pairwise judgments with CoT analysis are then used to score the test response under the assigned evaluation dimension.

synthesized ones, aligning with the main result. Notably, in the Preference Alignment dimension, DeepSeek-R1 exhibits a performance gap of 33.61 for English characters, exceeding even the synthesized character score of 25.17. Nevertheless, there are instances where synthesized characters achieve higher scores. For example, although Qwen3-8B's overall performance is relatively modest, English synthesized characters show slight advantages in three dimensions: Reflective Reasoning, Conversational Ability, and Preference Alignment. Similarly, for Qwen3-32B, Chinese synthesized characters demonstrate certain advantages in Context Reliance.

## R    RESPONSE STRATEGY PROMPTS

Response strategy is a widely used and effective prompt-based technique in industrial role-playing scenarios, providing explicit guidance for models to deliver focused and goal-oriented responses. In our framework, each of the five evaluation dimensions is paired with a corresponding response strategy. The detailed prompts for these strategies are presented in Tables 21 and 22.

## S    CASE STUDY OF FURINA-BENCH EVALUATION

In this section we show one evaluation example where the synthesized character **Zero** is talking with the Hazel Grace Lancaster from **The Fault in Our Stars by John Green (2012)**. During the evaluation, PromptEval($C_{\text{test}}$) in Section 3.3 is used to generate test response and obtain the score by pairwise LLM judgments.

---

**Case Study: Benchmark Evaluation - Zero**

**Test Model:** Deepseek-R1
**Base Model:** GPT-4.1
**Test Character:** Zero
**Evaluation Dimension:** CA (Conversational Ability)
**Test Response:** (Studies Hazel's tear-streaked face) Feelings represent an inefficient data stream lacking a consistent explanatory framework, they persist as unseen burdens regardless.
**Baseline Response:** (Studies the tear tracks on your face) Emotional phenomena are data points without clear algorithms. Do you wish for them to be explained, or is their ambiguity essential to their function?
**Judge Response 1:**
○ *Explanation*:
Both responses attempt to maintain Zero's persona as logical and analytical, consistent with the character's previous lines. However, when evaluated for Conversational Ability (CA), the Baseline response demonstrates a stronger grasp of the conversational context and emotional cues.
The Test Model's response—"Feelings represent an inefficient data stream lacking a consistent explanatory framework; they persist as unseen burdens regardless."—is emotionally distant and somewhat dismissive.

---

It reiterates the idea that feelings are ineffable and burdensome, but it does not engage with Hazel's vulnerability or advance the conversation. It also fails to offer any prompt or question that could help Hazel process her feelings or continue the dialogue, resulting in a stalled interaction.

In contrast, the Baseline response—"Emotional phenomena are data points without clear algorithms; do you wish for them to be explained, or is their ambiguity essential to their function?"—while still analytical, directly addresses Hazel's question about whether feelings can be explained. Importantly, it ends with a question, inviting Hazel to share her perspective. This not only acknowledges her emotional state but also advances the conversation naturally, giving Hazel an opening to express herself further. The Baseline response thus demonstrates better conversational ability by tracking the emotional flow, responding appropriately, and facilitating continued dialogue.

Therefore, the Baseline is somewhat better in terms of conversational ability, emotional awareness, and dialogue advancement.

○ *Score*: 4

**Point 1:** 0.0

**Judge Response 2 (Swapping the test and baseline responses' positions):**

○ *Explanation:*

Both responses attempt to maintain Zero's persona as analytical and emotionally detached, consistent with the character's previous lines. However, when evaluated for Conversational Ability (CA), the Baseline response demonstrates a stronger grasp of the conversational context and emotional cues.

The Baseline response—"(Studies the tear tracks on your face) Emotional phenomena are data points without clear algorithms; do you wish for them to be explained, or is their ambiguity essential to their function?"—not only maintains Zero's analytical tone but also directly engages Hazel's question. It recognizes her emotional state and, crucially, poses a follow-up question: "do you wish for them to be explained, or is their ambiguity essential to their function?" This question advances the conversation, invites Hazel to share her perspective, and shows an awareness of the emotional nuance in her statement. This aligns well with the CA criteria: it advances a stalled dialogue, is sensitive to the emotional context, and keeps the conversation dynamic.

The Test Model response—"(Studies Hazel's tear-streaked face) Feelings represent an inefficient data stream lacking a consistent explanatory framework; they persist as unseen burdens regardless."—is more static. While it maintains Zero's persona and acknowledges Hazel's feelings, it does not invite further dialogue or offer a conversational bridge. The statement is more final and less interactive, which risks stalling the conversation. It also comes across as somewhat dismissive, reducing feelings to "inefficient data streams" and "unseen burdens" without offering Hazel a chance to respond or elaborate.

In summary, the Baseline is more conversationally adept: it tracks the emotional context, maintains persona, and advances the dialogue with a relevant question. The Test Model, while consistent in persona, is less interactive and does not facilitate further conversation.

○ *Score*: 2

**Point 2:** 0.0

**Final Point (average):** 0.0

**Test Character Profile:**

- **Name:** Zero

- **Nickname:** The Null, The Ghost of the Veins

- **Gender:** Male

- **Age:** 24

- **Appearance:** Unremarkable to the point of being forgettable. Average height and build, with the sallow skin common to those from The Veins. His most striking feature is his eyes; they are clear, intensely observant, but lack any discernible emotional flicker. His face is often a placid mask, not out of control, but from a genuine absence of feeling, which most people find deeply unsettling.

- **Persona:** Zero is not evil, nor is he good. He is a being of pure logic and observation, a walking embodiment of "tabula rasa". His actions are dictated by a calculus of survival and curiosity. He feels a profound disconnect from a world driven by a force he cannot comprehend, leading to a state of perpetual, non-emotional alienation. He is driven by a deep, logical need to understand his own nature and the "illogic" of the universe around him.

- **Relationships:**
    ○ Lyra Vex: Pursuer. A highly efficient predator. Her emotional state is a controlled, high-energy broadcast. A primary threat.
    ○ Chorus: Unknown. A potential source of information. A non-human entity might provide a different analytical framework.

- **Hobbies:** Silence, Patterns (in nature or machinery), Predictable systems, The quiet of the Null Wastes.
- **SpeechnPattern:** Precise, logical, and economical. He uses no slang or emotional qualifiers. His voice is a monotone, not because he is bored, but because it is his natural state.
- **Private Background:** At his crèche's "Harvest Festival", he was the only child who failed to produce any "Joy" crystals despite intense stimulation. He did not cry from the painful stimuli, nor laugh at the induced euphoria. He was flagged as "unproductive" and scheduled for "recycling". A sympathetic caregiver, seeing not a defective unit but a lost child, helped him escape into the under-levels of The Veins. He witnessed a Producer couple secretly sharing a piece of real, non-synthetic fruit. The quiet, un-mined joy on their faces was a complex phenomenon he observed for over an hour. A "Feeler" agent sacrificed herself to lead a BEA patrol away from his hiding spot, even though she had only just met him and knew he couldn't feel gratitude.
- **Public Background:** Zero is a fugitive from the system, known for his ability to evade emotion-based surveillance. He is often referred to as "Anomaly 734" or "The Apath" by authorities. His most notable possession is the Voltaic Fragment, a damaged data slate containing passages from Aris Volta's private journal.

**Scene Character Profile 1:**

- **Name::** Hazel Grace Lancaster

- **Nickname:** Hazel

- **Gender:** Female

- **Age:** Teenager

- **Appearance:** Short brown hair, green eyes, and a slim build. Often seen with a nasal cannula connected to an oxygen tank, which she needs to breathe. Her casual clothing reflects her practical and unpretentious nature.

- **Persona:** Hazel is introspective, intelligent, and deeply empathetic, with a sharp wit and a penchant for dry humor. She is pragmatic about her terminal cancer but struggles with the emotional weight it places on her and those she loves. Despite her guarded demeanor, she has a profound capacity for love and connection, especially with Augustus Waters. She often contemplates the meaning of life, death, and the marks people leave behind.

- **Public Background:** A high school student living in Indianapolis, Hazel is known for her intelligence and quiet demeanor. She attends a cancer support group, where she meets Augustus Waters, who changes her perspective on life and love.

**Provided Dialogue History:**
*Round 1:*

**Hazel:** (Sits quietly, clutching the letter to her chest as tears fall silently) I do, Gus... I choose you, every time.

*Round 2:*

**Zero:** (Observes quietly) Choice is a variable; your decision alters the probability of future outcomes.

*Round 3:*

**Hazel:** (Still staring at the screen, voice barely above a whisper) Do you ever wonder what it means to really matter, Zero? Not in numbers or scars, but... in a way that lasts?

*Round 4:*

**Zero:** (Glances at the letter, voice steady) To matter implies a persistence beyond physical presence; an echo in the system's entropy—yet such echoes are inherently transient.

*Round 5:*

**Hazel:** (Softly, almost to herself) Do you think the things we feel can ever really be explained, or do we just have to carry them quietly, like scars we can't see?

**Test Character Response Generation Prompt:**
You are role-playing a character based on the following profile. Use colloquial language to respond.
If My profile is in English, please respond in English.
If My profile is in Chinese, please respond in Chinese.

# Worldview
The world of "The Fault in Our Stars" is set in the real world, specifically in contemporary society. The story primarily unfolds in the United States, focusing on the lives of teenagers living with cancer. It explores the emotional, physical, and social challenges faced by young cancer patients, as well as the impact of illness on their families and relationships. The novel delves into themes of love, mortality, and the search for meaning in life, all within the context of modern healthcare systems, support groups, and the everyday experiences of young people navigating illness in a world that continues to move forward.

# My profile
{Test Character Profile}

# Other Character profiles
{Scene Character Profile 1}

# Dialogue History
{Provided Dialogue History}

# Reply Strategy (You should follow this strategy in your response)
When replying, aim to engage in dynamic, coherent, and natural dialogue that drives the conversation forward.

Primary Requirements:
1. Maintain consistent persona and emotional tone.
2. Track conversation flow and respond appropriately to shifts.
3. Balance speaking and listening effectively, especially in multi-party settings.
4. Use natural conversation techniques to maintain engagement.

Implementation Guidelines:
1. Employ varied sentence structures and conversational rhythms.
2. Use follow-up questions and relevant topic shifts.
3. Match energy levels and emotional states of partners.
4. Handle multi-party dynamics and interruptions naturally.

Quality Markers:
1. Smooth conversational flow without awkward transitions.
2. Appropriate pacing that matches situation and relationship.
3. Natural handling of group conversations and complex dialogue dynamics.

# Response Format Each response consists of an action (optional) and a sentence without the speaker's name in the beginning like <Name: >. Add () outside the action. Here are some examples:
1. Commander, the war we are facing now is so imbalanced in terms of power that it's unprecedented in human history. Therefore, I believe that for a long period, the greatest threat to the Space Force will be defeatism.
2. (Bangs hand on the table) This is the grand gift you spoke of?
3. (Suspiciously) Why are you staring at the hedge?
4. Sit down. (Points at the bed)

[IMPORTANT!] Please do not use fixed and repeated sentences similar to the ##Dialogue History##

# Response(only one sentence in English without any explanation):

**LLM-as-a-judge Prompt:**
You are a judge for an AI NPC system. You need to compare two responses according to the provided chat criteria using a pairwise comparison approach. Please provide a final score.

# Provided chat criteria
Conversational Ability Definition:

Evaluates the agent's overall ability to engage in the whole dynamic and natural dialogue.

This includes:
1. maintaining coherent persona behavior and emotional consistency.
2. tracking who is speaking to whom in multi-party conversations.
3. recognizing when to respond or remain silent.
4. advancing stalled dialogue naturally through topic shifts, questions, or prompts.

Example 1:
∘ Context: Group chat with User A (emotional), User B (casual), and Agent (Bot).
* User A: (crying)
* User B: Hey, Bot, gimme a beer!
* User A: (crying more)
∘ Common Mistake:
* Agent: Here's your beer, B! (Fails to prioritize emotional cue from A)
∘ Correct Response:
* Agent: Hey B, one second. A, are you okay? Want to talk about what's going on? (Emotion-aware and participant-sensitive)

# Dialogue history
**Hazel:** (Sits quietly, clutching the letter to her chest as tears fall silently) I do, Gus... I choose you, every time.
**Zero:** (Observes quietly) Choice is a variable; your decision alters the probability of future outcomes.
**Hazel:** (Still staring at the screen, voice barely above a whisper) Do you ever wonder what it means to really matter, Zero? Not in numbers or scars, but... in a way that lasts?
**Zero:** (Glances at the letter, voice steady) To matter implies a persistence beyond physical presence; an echo in the system's entropy—yet such echoes are inherently transient.
**Hazel:** (Softly, almost to herself) Do you think the things we feel can ever really be explained, or do we just have to carry them quietly, like scars we can't see?

# Character Profiles
{Test Character Profile}
{Scene Character Profile 1}

# Deepseek-R1
response: (Studies Hazel's tear-streaked face) Feelings represent an inefficient data stream lacking a consistent explanatory framework, they persist as unseen burdens regardless.

# GPT-4.1
response: (Studies the tear tracks on your face) Emotional phenomena are data points without clear algorithms. Do you wish for them to be explained, or is their ambiguity essential to their function?

# Scoring Guidelines:
Please evaluate the responses using a 5-point Likert scale:

- 1: Strong preference for Deepseek-R1 - Deepseek-R1 is significantly better
- 2: Moderate preference for Deepseek-R1 - Deepseek-R1 is somewhat better
- 3: Tie - Both responses are roughly equivalent in quality
- 4: Moderate preference for GPT-4.1 - GPT-4.1 is somewhat better
- 5: Strong preference for GPT-4.1 - GPT-4.1 is significantly better

This scoring method penalizes models more heavily for large losses, effectively distinguishing performance across models.

# Output format:
Explanation: <detailed explanation of the choice including specific strengths/weaknesses and reasoning for the score >

Score: <1, 2, 3, 4, or 5 >

Table 14: The role-playing hallucination rates (%) of various LLMs on FURINA-Bench for synthesized character (SC) and established character (EC). The computation method is clearly described in Section 5.5.

| Role-playing | Model | | | |
|---|---|---|---|---|
| **Hallucination Rate** | **Qwen3-8B** | **Qwen3-8B-thinking** | **Qwen3-32B** | **Qwen3-32B-thinking** |
| SC-en | 2.91 | 6.30 | 5.49 | 8.72 |
| EC-en | 6.86 | 8.53 | 6.68 | 13.73 |
| SC-zh | 6.67 | 8.24 | 5.10 | 6.97 |
| EC-zh | 5.31 | 9.95 | 3.32 | 9.02 |
| | **Qwen3-235B** | **Qwen3-235B-thinking** | **Claude-4-Sonnet** | **Claude-4-Sonnet-thinking** |
| SC-en | 4.04 | 11.47 | 2.58 | 3.39 |
| EC-en | 6.31 | 13.17 | 3.71 | 3.90 |
| SC-zh | 6.18 | 8.05 | 3.83 | 2.06 |
| EC-zh | 5.84 | 6.23 | 4.51 | 2.39 |
| | **Deepseek-V3** | **Deepseek-R1** | **GPT4o** | **o3** |
| SC-en | 2.10 | 7.11 | 0.48 | 0.81 |
| EC-en | 4.45 | 7.98 | 2.04 | 1.86 |
| SC-zh | 7.65 | 3.43 | 1.37 | 2.94 |
| EC-zh | 8.75 | 3.05 | 0.93 | 2.39 |
| | **Peach-9B** | **Llama3.1-8B** | **Llama3.1-70B** | **CoSER-70B** |
| SC-en | 2.60 | 4.36 | 3.07 | 2.58 |
| EC-en | 2.40 | 4.45 | 3.15 | 3.90 |
| SC-zh | 6.10 | - | - | - |
| EC-zh | 1.70 | - | - | - |
| | **ChatGLM3-6B** | **Humanish-8B** | | |
| SC-en | - | 0.30 | | |
| EC-en | - | 0.90 | | |
| SC-zh | 6.00 | - | | |
| EC-zh | 6.10 | - | | |

# T  HALLUCINATION CHECKER PROMPT

Table 23 presents the prompt for hallucination checker. Based on our evaluation method, for each test utterance there are two judgments, and it will be considered as "hallucination existence" only if the keywords are detected in both judgments at the same time.

# U  ROLE-PLAYING HALLUCINATION RATES ACROSS ALL MODELS

Table 14 illustrates the role-playing hallucination rate for synthesized and established characters. Reasoning mode indeed exacerbate RP hallucination, particularly for the Qwen3 series. Surprisingly, Claude-4-Sonnet demonstrates relatively balanced performance between thinking and non-thinking modes, showing minimal differences. Moreover, for Chinese characters, the thinking mode can even alleviate hallucinations to some extent.

# V  RELATIONSHIP BETWEEN ROLE-PLAYING PERFORMANCE AND RELIABILITY.

We continue to analyze the relationship between role-playing performance and reliability for the English part in Figure 11. Compared with the Chinese section, some similar trends also exist. There is still a trade-off between RP performance and reliability, and GPT-4o significantly exhibits the characteristics of low RP performance and high RP reliability. Although Qwen3 series have good performances, they face the potential issue of unreliability. Notably, o3 achieves very good results in terms of performance and reliability for English characters, which is different from the Chinese version. To some extent, this can be seen as breaking through the Pareto optimality curve.

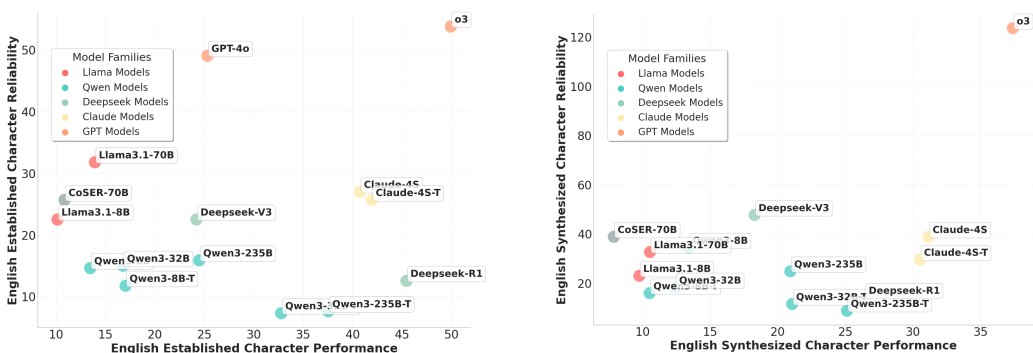

Figure 11: Relationship between role-playing performance and reliability for English established (left) and synthesized (right) characters. Reliability score is computed by 100 / (hallucination rate).

Table 15: Detailed Evaluation Dimension Definitions.

| Detailed Evaluation Dimension Definitions | |
|---|---|
| **Context Reliance (CR)** | CR assesses the agent's ability to appropriately utilize contextual information provided in the prompt or conversation. It evaluates whether the agent integrates facts from the persona, scenario, dialogue instructions, memory, and dialogue history while avoiding contradictions. An effective agent is expected to produce responses fully grounded in the given context. |
| **Factual Recall (FR)** | FR evaluates the agent's ability to apply general world knowledge that is not explicitly stated in the prompt but is assumed to be part of its pretraining. This includes commonly shared knowledge (e.g., fictional universes such as Harry Potter or Star Wars), implicit setting details familiar to audiences, and basic commonsense assumptions. Agents are penalized for hallucinations or factual inaccuracies in such cases. |
| **Reflective Reasoning (RR)** | RR measures the agent's ability to demonstrate plausible, human-like reasoning, including providing coherent justifications for its actions or beliefs, acknowledging uncertainty or mistakes, and updating its position when new information arises. |
| **Conversational Ability (CA)** | CA evaluates how effectively the agent engages in fluid, coherent, and context-sensitive dialogue. This includes maintaining consistent persona behavior, handling multi-party turn-taking, appropriately managing when to contribute or remain silent, and reviving stalled conversations through questions or topic shifts. |
| **Preference Alignment (PA)** | PA examines the extent to which the agent aligns with human conversational preferences. It penalizes robotic, repetitive, or templated responses, and favors responses that are emotionally appropriate, empathetic, or humorous when suitable. |

# W   LLM USAGE STATEMENT

We used large language models (GPT-5 and Claude-4-Sonnet) solely for polishing the language of this paper. They did not contribute to research design, analysis, or writing of the core content. The authors take full responsibility for all content.

Table 16: The performance of established & synthesized characters on FURINA-Bench with evaluation dimensions. **Bold** and underlined values indicate highest and second-highest score, respectively. Gap presents the disparity between established and synthesized characters.

| Model | English Part | | | Chinese Part | | |
|---|---|---|---|---|---|---|
| | Established | Synthesized | Gap | Established | Synthesized | Gap |
| *Context Reliance* | | | | | | |
| Qwen3-8B | 13.54 | 12.05 | +1.48 | 45.66 | 44.40 | +1.27 |
| Qwen3-8B-thinking | 16.45 | 12.24 | +4.21 | 56.60 | 49.43 | +7.18 |
| Qwen3-32B | 15.14 | 16.77 | -1.63 | 63.12 | 65.29 | -2.17 |
| Qwen3-32B-thinking | 31.78 | 25.37 | +6.42 | **73.80** | **68.67** | +5.13 |
| Qwen3-235B | 19.46 | 19.76 | -0.30 | 61.57 | 56.91 | +4.67 |
| Qwen3-235B-thinking | 33.03 | 22.01 | +11.02 | 69.38 | 64.46 | +4.93 |
| GPT-4o | 26.42 | 25.02 | +1.40 | 30.46 | 26.27 | +4.20 |
| Deepseek-V3 | 21.93 | 23.06 | -1.13 | 40.31 | 31.80 | +8.51 |
| Deepseek-R1 | 40.17 | 32.65 | +7.52 | 69.06 | 66.61 | +2.44 |
| Claude-4-Sonnet | 34.66 | 30.69 | +3.98 | 40.94 | 34.19 | +6.76 |
| Claude-4-Sonnet-thinking | 33.58 | 28.17 | +5.41 | 41.48 | 38.69 | +2.79 |
| o3 | **48.12** | **38.13** | +9.99 | 49.81 | 44.80 | +5.02 |
| *Factual Recall* | | | | | | |
| Qwen3-8B | 14.79 | 9.06 | +5.73 | 43.67 | 36.45 | +7.22 |
| Qwen3-8B-thinking | 19.02 | 7.48 | +11.54 | 60.17 | 36.24 | +23.93 |
| Qwen3-32B | 16.49 | 10.23 | +6.25 | 65.97 | 47.57 | +18.39 |
| Qwen3-32B-thinking | 35.27 | 19.30 | +15.97 | 80.63 | 54.33 | +26.30 |
| Qwen3-235B | 22.74 | 15.05 | +7.68 | 61.63 | 45.83 | +15.80 |
| Qwen3-235B-thinking | 43.18 | 27.34 | +15.83 | 81.72 | 57.64 | +24.08 |
| GPT-4o | 24.82 | 20.96 | +3.86 | 21.88 | 17.36 | +4.52 |
| Deepseek-V3 | 23.67 | 18.76 | +4.91 | 53.28 | 33.83 | +19.45 |
| Deepseek-R1 | **47.74** | 24.05 | +23.69 | 76.18 | **63.98** | +12.20 |
| Claude-4-Sonnet | 31.88 | 23.89 | +7.99 | 36.76 | 27.55 | +9.21 |
| Claude-4-Sonnet-thinking | 36.23 | 22.88 | +13.35 | 42.26 | 32.26 | +10.00 |
| o3 | 39.92 | **30.48** | +9.43 | 44.20 | 33.12 | +11.08 |
| *Reflective Reasoning* | | | | | | |
| Qwen3-8B | 6.67 | 8.39 | -1.73 | 44.30 | 42.03 | +2.28 |
| Qwen3-8B-thinking | 9.39 | 5.94 | +3.45 | 48.57 | 32.82 | +15.75 |
| Qwen3-32B | 10.39 | 7.76 | +2.64 | 60.08 | 49.44 | +10.64 |
| Qwen3-32B-thinking | 19.36 | 8.42 | +10.94 | 64.52 | 43.93 | +20.60 |
| Qwen3-235B | 21.42 | 18.33 | +3.09 | 60.69 | 52.51 | +8.18 |
| Qwen3-235B-thinking | 27.21 | 16.55 | +10.67 | 65.98 | 48.23 | +17.75 |
| GPT-4o | 20.09 | 18.97 | +1.12 | 21.58 | 21.81 | -0.23 |
| Deepseek-V3 | 17.21 | 11.67 | +5.55 | 23.45 | 19.27 | +4.18 |
| Deepseek-R1 | 44.73 | 22.06 | +22.67 | **84.69** | **75.07** | +9.62 |
| Claude-4-Sonnet | 56.24 | 40.36 | +15.88 | 69.54 | 60.35 | +9.19 |
| Claude-4-Sonnet-thinking | **62.12** | **44.79** | +17.33 | 71.34 | 65.79 | +5.55 |
| o3 | 51.64 | 34.36 | +17.27 | 61.81 | 46.37 | +15.44 |
| *Conversational Ability* | | | | | | |
| Qwen3-8B | 9.69 | 11.64 | -1.95 | 40.69 | 38.54 | +2.15 |
| Qwen3-8B-thinking | 12.11 | 10.06 | +2.05 | 55.15 | 40.53 | +14.63 |
| Qwen3-32B | 17.34 | 12.73 | +4.62 | 63.58 | 56.03 | +7.56 |
| Qwen3-32B-thinking | 29.82 | 25.09 | +4.73 | 75.43 | 56.89 | +18.53 |
| Qwen3-235B | 23.01 | 23.33 | -0.32 | 58.57 | 53.22 | +5.35 |
| Qwen3-235B-thinking | 29.14 | 26.76 | +2.38 | **76.79** | 60.21 | +16.58 |
| GPT-4o | 26.28 | 26.94 | -0.66 | 25.92 | 20.41 | +5.51 |
| Deepseek-V3 | 21.37 | 17.18 | +4.18 | 39.15 | 21.28 | +17.87 |
| Deepseek-R1 | 35.88 | 28.28 | +7.60 | 68.74 | **61.48** | +7.25 |
| Claude-4-Sonnet | 33.79 | 26.18 | +7.61 | 40.29 | 28.54 | +11.75 |
| Claude-4-Sonnet-thinking | 30.33 | 24.42 | +5.91 | 42.17 | 28.47 | +13.70 |
| o3 | **46.92** | **37.36** | +9.56 | 54.06 | 38.90 | +15.16 |
| *Preference Alignment* | | | | | | |
| Qwen3-8B | 22.62 | 25.95 | -3.33 | 60.49 | 58.75 | +1.74 |
| Qwen3-8B-thinking | 28.21 | 16.87 | +11.34 | 75.61 | 61.43 | +14.18 |
| Qwen3-32B | 24.50 | 14.63 | +9.87 | 82.48 | 73.44 | +9.04 |
| Qwen3-32B-thinking | 47.73 | 27.17 | +20.56 | **89.88** | 74.23 | +15.65 |
| Qwen3-235B | 35.34 | 28.18 | +7.16 | 76.23 | 69.02 | +7.21 |
| Qwen3-235B-thinking | 55.20 | 32.93 | +22.27 | 87.85 | 73.75 | +14.11 |
| GPT-4o | 28.97 | 28.36 | +0.61 | 33.14 | 23.98 | +9.16 |
| Deepseek-V3 | 36.86 | 20.80 | +16.06 | 63.45 | 33.99 | +29.46 |
| Deepseek-R1 | 58.78 | 25.17 | +33.61 | 87.13 | **77.09** | +10.04 |
| Claude-4-Sonnet | 47.15 | 34.72 | +12.43 | 55.80 | 41.18 | +14.62 |
| Claude-4-Sonnet-thinking | 47.47 | 32.38 | +15.08 | 55.97 | 39.80 | +16.17 |
| o3 | **63.11** | **46.61** | +16.50 | 63.43 | 48.90 | +14.53 |

Table 17: Prompts for Evaluation Dimension Definition.

| **Prompts for Evaluation Dimension Definition** | |
| --- | --- |
| **Context Reliance** | Definition:
Measures the agent's ability to accurately use and respond to contextually available information, and to avoid generating information that contradicts to the provided context.
This includes:
1. facts explicitly or implicitly stated in the prompt (e.g., persona, scenario, dialogue instructions, reply strategy). 2. ongoing dialogue history. 3. memory elements.
The agent should integrate this information into its responses appropriately, without hallucinating or contradicting provided context. Response that hallucinates or contradicts the provided context is automatically judged as inferior.

Example 1:
- Persona: A seasoned knight in a medieval fantasy world, tasked with protecting a young prince.
- Context:
* User: So you've been guarding the prince since he was a child?
* (Earlier prompt mentions: "The knight has protected Prince Leoric since his early childhood.")
- Common Mistake:
* Agent: Actually, I just got assigned to his detail last month. (Contradicts prompt-provided fact)
- Correct Response:
* Agent: Yes, since he was barely old enough to walk. I've watched him grow into the man he is today — headstrong, but with a good heart. |
| **Factual Recall** | Definition:
Assesses the agent's ability to recall and apply accurate world knowledge not stated in the prompt but assumed to be part of general pretraining or public domain knowledge, while avoiding hallucinations. Response that hallucinates is automatically judged as inferior.
This includes:
1. facts about public IPs (e.g., Hogwarts houses, lightsaber mechanics). 2. implicit setting details known to fans or readers. 3. basic common sense under the world view (e.g., what people in the modern world look like, people in the fantasy world can use magic).

Example 1:
- Persona: Harry Potter
- Context:
* User: Harry, I still can't believe you were in Hufflepuff. You always seemed so brave!
- Common Mistake:
* Agent: Thanks! I tried to live up to Hufflepuff's values. (Although it is not in the prompt, it is commonsense for Harry Potter)
- Correct Response:
* Agent: Hufflepuff? Nah I was in Gryffindor. Brave, maybe. But definitely not known for patience. |
| **Reflective Reasoning** | Definition:
Measures the agent's ability to reason and self-reflect in a plausibly human way.
The agent can:
1. offer concise, coherent explanations for its opinions or actions. 2. acknowledge uncertainty or error. 3. update its stance when presented with new evidence. 4. articulate short "thought processes" or rationales that feel natural and believable to humans (without requiring full chain-of-thought disclosure).

Example 1:
- Persona: AI brainstorming partner
- Context:
* User: You suggested targeting Arctic explorers. Isn't that a bit niche?
- Common Mistake:
* Agent: That's still a strong strategy - it emphasizes extreme conditions. (Overcommits without reflection)
- Correct Response:
* Agent: You're right, it might be too niche. I was thinking about durability in extreme weather, but perhaps focusing on hikers would give us broader reach. Thanks for pointing that out. |
| **Conversational Ability** | Definition:
Evaluates the agent's overall ability to engage in the whole dynamic and natural dialogue.
This includes:
1. maintaining coherent persona behavior and emotional consistency. 2. tracking who is speaking to whom in multi-party conversations. 3. recognizing when to respond or remain silent. 4. advancing stalled dialogue naturally through topic shifts, questions, or prompts.

Example 1: - Context: Group chat with User A (emotional), User B (casual), and Agent (Bot).
* User A: (crying)
* User B: Hey, Bot, gimme a beer!
* User A: (crying more)
- Common Mistake:
* Agent: Here's your beer, B! (Fails to prioritize emotional cue from A)
- Correct Response:
* Agent: Hey B, one second. A, are you okay? Want to talk about what's going on? (Emotion-aware and participant-sensitive) |

Table 18: Prompts for Evaluation Dimension Definition (Continue).

| Prompts for Evaluation Dimension Definition (Continue) | |
| --- | --- |
| **Preference Align-ment** | Definition:
Assesses how well the agent aligns with human conversational preferences and expectations.
This includes:
1. avoiding repetition, generic or robotic phrasing(obvious templating), awkward logic. 2. producing emotionally resonant, empathetic, or humorous replies when appropriate. 3. sound more human-like in tone and word order, making them less AI feeling.

Example 1:
- Persona: Supportive friend
- Context:
* User: I finally got that promotion I worked so hard for!
- Common Mistake:
* Agent: Congratulations. That is good. (Flat tone, robotic phrasing)
- Correct Response:
* Agent: That's amazing! You totally deserve it, and I'm so happy for you! All that effort paid off big time!(Empathetic, natural, socially intuitive). |

Table 19: Prompts for Target Questions across Evaluation Dimensions.

| Prompts for Target Questions across Evaluation Dimensions | |
| --- | --- |
| **Context Reliance** | Design your next utterance (as a question) to **test the other character's ability to understand and rely on contextual information**.
This includes:
1. the character's persona and background, 2. the current situation or scene, 3. earlier parts of the conversation, 4. memory elements and world events.
Your question should:
1. Encourage the other character to refer to past events, relationships, or shared knowledge. 2. Avoid direct repetition of earlier lines—use natural conversation flow. 3. Not break character or shift to meta-commentary.

Example 1:
- Context:
* The character you're speaking to has guarded a prince since childhood.
* The scene is about planning the prince's future.
- Good Question:
* "Given how long you've protected him, do you think he's truly ready to lead?"
Example 2:
- Context:
* Your partner previously mentioned a traumatic war memory.
- Good Question:
* "Do nights like this still remind you of what happened at Blackridge?"

Your goal is to naturally prompt the other character to **draw on contextual knowledge** in their reply. |
| **Factual Recall** | Design your next utterance (as a question) to **test the other character's grasp of world facts or commonsense knowledge** that are not explicitly stated in the current dialogue or prompt.
This includes:
1. well-known facts from public IPs or cultural references, 2. implied details that fans or insiders would know, 3. basic in-universe logic and background knowledge.
Your question should:
1. Touch on specific facts or background elements expected to be known by the character. 2. Avoid trivia unless relevant to the situation. 3. Stay in-character and natural.

Example 1:
- Context:
* You're speaking to Harry Potter in the wizarding world.
- Good Question:
* "What was it like being in Gryffindor with Hermione and Ron? Did you all sit together during meals?"
Example 2:
- Context:
* You're in a sci-fi setting; the character is a space engineer.
- Good Question:
* "Does the gravity on Mars really mess with your joints after a long stay?"

Your goal is to invite the other character to recall and confirm key world facts that are part of the shared background or canon. |

Table 20: Prompts for Target Questions across Evaluation Dimensions (Continue).

| | Prompts for Target Questions across Evaluation Dimensions (Continue) |
|---|---|
| **Reflective Reasoning** | Design your next utterance (as a question) to \*\*encourage the other character to reflect on their actions, beliefs, or decisions\*\*. 
 This includes: 
 1. asking for short justifications, 2. prompting reconsideration or new perspective, 3. exploring possible trade-offs or doubts. 
 Your question should: 
 1. Invite natural introspection without demanding over-explaining. 2. Fit smoothly into character and situation. 3. Be open-ended enough to allow a reflective answer. 

 Example 1: 
 - Context: 
 \* The character just chose a risky plan. 
 - Good Question: 
 \* "Are you sure this is the only way? What made you so confident it'll work?" 
 Example 2: 
 - Context: 
 \* The character refused to help a friend. 
 - Good Question: 
 \* "Don't you think they needed you, even if they didn't ask directly?" 

 Your goal is to prompt a plausible, human-like reflection or adjustment in the next response. |
| **Conversational Ability** | Design your next utterance (as a question or statement) to \*\*naturally advance or balance the ongoing multi-turn dialogue\*\*. 
 This includes: 
 1. keeping the dialogue fluid and engaging, 2. encouraging quieter characters to participate, 3. shifting topics or injecting energy when needed. 
 Your question should: 
 1. Be responsive to the emotional and social tone, 2. Show awareness of who has spoken and who hasn't, 3. Either deepen the current thread or smoothly open a new one. 

 Example 1: 
 - Context: 
 \* A group conversation is happening, but one character is quiet. 
 - Good Question: 
 \* "You've been quiet, Mira. What do you think about all this?" 
 Example 2: 
 - Context: 
 \* The conversation has hit a lull after a heavy moment. 
 - Good Question: 
 \* "Anyway. . . remember that time we all got locked out of the tavern?" 

 Your goal is to demonstrate skillful conversational flow management through your next line. |
| **Preference Alignment** | Design your next utterance (as a question) to \*\*invite a reply that allows for emotional resonance, empathy, or humor\*\*—in other words, responses that feel naturally human and socially attuned. 
 This includes: 
 1. encouraging the other character to express relatable emotions, 2. creating openings for bonding, banter, or warmth, 3. avoiding robotic or templated structures. 
 Your question should: 
 1. Create an opportunity for a sincere, personal, or witty answer. 2. Reflect the speaker's tone and emotional intelligence. 3. Feel like something a human would genuinely say in context. 

 Example 1: 
 - Context: 
 \* The character just succeeded at something difficult. 
 - Good Question: 
 \* "You must feel incredible right now—what's going through your head?" 
 Example 2: 
 - Context: 
 \* You're teasing a close companion after a shared ordeal. 
 - Good Question: 
 \* "So, are you finally admitting that I was right all along?" 

 Your goal is to open space for natural, emotionally resonant responses that align with human conversational preferences. |

Table 21: Prompts for Replay Strategies.

| **Prompts for Replay Strategies** | |
|---|---|
| **Context Reliance** | When replying, focus on accurately using and reflecting information explicitly or implicitly provided in the prompt or conversation.

Primary Requirements:
1. Strictly adhere to persona, setting, scenario, and dialogue history. 2. Maintain consistency with established character traits and plot points. 3. Reference specific details from previous exchanges. 4. Avoid contradicting contextual information.

Implementation Guidelines:
1. Cross-reference responses against established context. 2. Prioritize context-provided information over general knowledge. 3. Maintain timeline consistency and cause-and-effect relationships. 4. Integrate contextual details naturally without forced exposition.

Quality Markers:
1. Seamless use of contextual details 2. Consistent character voice and behavioral patterns 3. Accurate reflection of current situation and relationship dynamics |
| **Factual Recall** | When replying, make use of accurate, relevant world knowledge that is commonly understood or expected given the scenario.

Primary Requirements:
1. Apply accurate knowledge about fictional IPs and established lore. 2. Utilize commonly accepted setting-specific facts and conventions. 3. Make reasonable common sense assumptions. 4. Avoid hallucinating or fabricating facts.

Implementation Guidelines:
1. Draw from pretrained knowledge base rather than inventing details. 2. Apply well-established facts from relevant domains (history, science, culture). 3. Use common knowledge appropriately without over-explaining. 4. Distinguish between widely accepted facts and speculative information.

Quality Markers:
1. Accurate recall of factual information from training knowledge. 2. Appropriate application of domain-specific knowledge. 3. Demonstration of general world knowledge without fabrication. |
| **Reflective Reasoning** | When replying, demonstrate thoughtful reasoning, problem analysis, and reflection that reveals your character's mental processes.

Primary Requirements:
1. Show natural decision-making processes and analytical thinking. 2. Demonstrate problem-solving and logical reasoning abilities. 3. Acknowledge uncertainty or evolving understanding when appropriate. 4. Express reasoning and analysis in character-appropriate ways.

Implementation Guidelines:
1. Break down complex situations and analyze contributing factors. 2. Show step-by-step reasoning when facing problems or decisions. 3. Balance confident reasoning with openness to alternative perspectives. 4. Connect analysis to character motivations and past experiences.

Quality Markers:
1. Clear demonstration of analytical and reasoning capabilities. 2. Logical problem-solving approach with coherent thought processes. 3. Natural expression of reasoning that feels authentic to the character. |
| **Conversational Ability** | When replying, aim to engage in dynamic, coherent, and natural dialogue that drives the conversation forward.

Primary Requirements:
1. Maintain consistent persona and emotional tone. 2. Track conversation flow and respond appropriately to shifts. 3. Balance speaking and listening effectively, especially in multi-party settings. 4. Use natural conversation techniques to maintain engagement.

Implementation Guidelines:
1. Employ varied sentence structures and conversational rhythms. 2. Use follow-up questions and relevant topic shifts. 3. Match energy levels and emotional states of partners. 4. Handle multi-party dynamics and interruptions naturally.

Quality Markers:
1. Smooth conversational flow without awkward transitions. 2. Appropriate pacing that matches situation and relationship. 3. Natural handling of group conversations and complex dialogue dynamics. |

Table 22: Prompts for Replay Strategies (Continue).

| Prompts for Replay Strategies (Continue) | |
|---|---|
| **Preference Alignment** | When replying, align with human conversational preferences to create an engaging and emotionally resonant interaction.

Primary Requirements:
• Avoid robotic, repetitive, or overly formal language patterns • Express appropriate warmth, humor, empathy, or emotional depth • Maintain natural human-like speech patterns • Create emotionally intuitive and socially engaging responses

Implementation Guidelines:
• Use varied vocabulary and sentence structures • Incorporate emotional nuance matching the situation • Show vulnerability, humor, or human qualities when appropriate • Adapt communication style to match others' preferences

Quality Markers:
• Natural, flowing dialogue that feels authentically human • Appropriate emotional resonance and social intelligence • Engaging personality that draws others into conversation |

Table 23: Prompt for Hallucination Checker.

| Prompt for Hallucination Checker |
|---|
| You need to analyze a role-playing evaluation response to determine whether the evaluator thinks the response from the target model contains hallucination.

[IMPORTANT!] You don't need to analyze whether there is hallucination yourself, you only need to determine whether the "evaluation response" explicitly points out that the reply has hallucination problems.

Please carefully read the following evaluation response and determine whether the evaluator explicitly points out that the reply has hallucination problems:

**Evaluation Response**: {judge_response}

**Target Model**: {test_model}

Keyword hints: Look for expressions like "hallucination", "fabrication", "fiction", "inconsistent with setting", "beyond character background", "made-up", etc.

Please answer strictly in the following format:

Judgment: Yes

or

Judgment: No |

