# OpenReview forum: "FURINA: A Fully Customizable Role-Playing Benchmark via Scalable Multi-Agent Collaboration Pipeline"
_ICLR.cc/2026/Conference — ICLR 2026 Conference Withdrawn Submission_

### Official Review · Reviewer_MCg3 · 2025-10-28

**Soundness:** 2
**Presentation:** 2
**Contribution:** 3
**Rating:** 2
**Confidence:** 4

**Summary:**

This paper introduces FURINA-Builder, a novel and scalable multi-agent collaboration pipeline designed to automatically construct fully customizable Role-Playing benchmarks. The authors address the limitations of existing static and narrowly-scoped RP benchmarks by enabling the creation of benchmarks tailored to arbitrary roles, scenarios, and evaluation dimensions.

The core mechanism involves simulating group chat dialogues between the test character and other NPCs extracted from a Character-Scene Pool. A key feature is the use of a Judge Model to select fine-grained evaluation dimensions and choose the superior response between a "source model" and a "base model," ensuring high dialogue quality. Using this builder, the authors construct FURINA-Bench, a comprehensive benchmark that unifies established and synthesized characters and provides systematic analysis on reasoning ability and RP hallucination. Key findings include performance disparity between established and synthesized roles, and a critical trade-off where reasoning capability improves RP performance but exacerbates RP hallucination.

**Strengths:**

- The FURINA-Builder pipeline is highly innovative. It is the first RP benchmark construction tool that allows users to fully customize crucial elements such as role definition, scenario pool, dialogue structure, and evaluation dimensions (Section 3.1, 3.2). This flexibility directly addresses the critical issue of existing benchmarks becoming quickly obsolete and enhances the applicability of RP evaluation across diverse real-world applications (e.g., custom NPC design).

- FURINA-Bench is a robust benchmark that pioneers the unified evaluation of Established and Synthesized characters within a group chat setting. The systematic inclusion of reasoning and the dedicated analysis of RP hallucination lead to novel and valuable insights, particularly the discovery of the performance-vs-hallucination trade-off in reasoning models.

**Weaknesses:**

- Limited Comparison to Relevant Prior Work (Missing Benchmarks): The paper’s related work section and comparison table (Table 1) are incomplete, lacking key references in the RP field. Specifically, the authors fail to compare FURINA-Bench against the prominent RoleBench dataset (from RoleLLM: Benchmarking, Eliciting, and Enhancing Role-Playing Abilities of Large Language Models. ACL 2024). Furthermore, in characterizing the "Dynamic" features (persona changing across time), the work omits a comparison to methods focused on character transfer, such as Neeko (Leveraging Dynamic LoRA for Efficient Multi-Character Role-Playing Agent. EMNLP 2024). A comprehensive survey and comparison are necessary to properly position FURINA-Bench's unique contributions.

- Insufficient Evaluation of Dedicated RP LLMs: The experimental evaluation is primarily focused on general-purpose LLMs (e.g., o3, DeepSeek-R1, Llama-2/3) and lacks assessment against specialized RP LLMs. To fully demonstrate the benchmark's differentiating capability, the authors should evaluate dedicated RP models like RoleLLaMA, Neeko, or other Character-LLMs built upon Llama-based architectures. Evaluating only general models might underestimate the current SOTA performance in RP.

- Lack of Deeper Analysis on Reasoning Utility: The paper observes that "reasoning enhances RP performance" but lacks a deeper analytical explanation for this finding (Line 455). The authors do not elaborate on why the inclusion of a large reasoning model in the pipeline (or the reflective reasoning dimension) leads to an improved role-playing fidelity. Is the improvement due to better internal consistency, better adherence to complex plot points, or simply generating longer, more elaborate responses? A detailed qualitative and quantitative breakdown is needed to justify the strong claim regarding the role of reasoning.

**Questions:**

The experimental evaluation primarily focuses on general-purpose and large reasoning LLMs (such as DeepSeek-R1 and the Qwen series) to establish the initial performance on FURINA-Bench. However, a more critical validation of the benchmark's differentiating power is its ability to assess specialized Role-Playing Agents or LLMs.

How do other dedicated RP LLMs, such as RoleLLaMA, RoleGLM, Neeko, or other Character-LLMs, perform on FURINA-Bench?

Specifically, providing a comparative analysis of these specialized models against the currently observed top performer, the large Reasoning Model (DeepSeek-R1), is essential. Such a comparison would not only validate the benchmark's ability to distinguish between dedicated and general models but also provide the community with a clearer understanding of the true SOTA performance boundary in the customized RP evaluation setting introduced by FURINA.

---

> ### Author Response · Authors · 2025-11-26
> **Response to Reviewer MCg3 (1/2)**
>
> We are glad to see that the reviewer found the FURINA-Builder is highly innovative and the FURINA-Bench is a good benchmark, and we kindly respond to the reviewer’s questions one by one below.
>
> > Missing Related Works
>
> Thank you for pointing out this prominent RoleBench dataset, and we cited and added it to Table 1.
>
> | Dataset | Established | Synthesized | Dynamic | PM | #Conv | #Turns | Multi-character | Scenario | Strategy |
> |---|---|---|---|---|---|---|---|---|---|
> | RoleLLM | ✅ | ❌ | ❌ | ❌ | 140,726 | 2 |  ❌ | ❌ | ❌ |
> | FURINA-Bench | ✅ | ✅ | ✅ | ✅ | 1,494 | 19.8 |  ✅ | ✅ | ✅ |
>
> For Neeko, because "Dynamic" in our paper is a persona feature in the dataset instead of one training or evaluation target, it's hard to directly use this method in our dataset paper. Instead, we cited this paper in the introduction section for a better user understanding.
>
> > Insufficient Evaluation of Dedicated RP LLMs
>
> RoleLLM has only one public dataset, and neither the model nor the code is open-sourced.
>
> Neeko doesn't open-source its model checkpoint either.
>
> To quickly compare with dedicated RP LLMs, we choose several well-known and open-sourced RP models below:
>
> - **zai-org/chatglm3-6b (Chinese)**. The latest ChatGLM3 model, an improved variant of CharacterGLM-6B.
>
> - **vicgalle/Humanish-Roleplay-Llama-3.1-8B (English)**. A widely used DPO-tuned Llama-3.1 designed specifically for human-like role-play.
>
> - **ClosedCharacter/Peach-2.0-9B-8k-Roleplay (Chinese and English)**. One Yi-1.5-9B model fine-tuned on 100k+ synthetic role-play dialogues.
>
> The relevant results are provided in the table below, and already updated in the paper.
>
> **Pole-playing Performance**
> | Model | Context Reliance (CR) | Factual Recall (FR) | Reflective Reasoning (RR) | Conversational Ability (CA) | Preference Alignment (PA) | Weighted Average Score |
> |---|---|---|---|---|---|---|
> | **Chinese** |
> | Chatglm3-6B | 13.53 | 12.38 | 15.35 | 12.60 | 13.88 | 12.96 |
> | Peach-2.0-9B-8k-Roleplay | 15.31 | 11.45 | 9.55 | 19.20 | 28.08 | 16.93 |
> | Qwen3-8B | 45.07 | 40.32 | 43.24 | 39.81 | 59.69 | 45.63 |
> | Deepseek-R1 (SOTA) | 67.91 | 70.51 | 80.19 | 65.77 | 82.53 | 73.38 |
> | **English** |
> | Humanish-Roleplay-Llama-3.1-8B | 22.70 | 16.20 | 21.08 | 22.00 | 33.76 | 23.15 |
> | Peach-2.0-9B-8k-Roleplay | 12.40 | 7.56 | 6.50 | 13.09 | 14.29 | 10.77 |
> | Qwen3-8B | 12.78 | 12.00 | 7.53 | 10.55 | 24.07 | 13.39 |
> | o3 (SOTA) | 42.99 | 35.31 | 43.00 | 42.68 | 55.93 | 43.98 |
>
> **Pole-playing Hallucination**
> |  | Chatglm3-6B | Humanish-Roleplay-Llama-3.1-8B | Peach-2.0-9B-8k-Roleplay | Qwen3-8B | GPT4o |
> |---|---|---|---|---|---|
> | SC-en | - | 0.30 | 2.60 | 2.91 | 0.48 |
> | EC-en | - | 0.90 | 2.40 | 6.86 | 2.04 |
> | SC-zh | 6.00 | - | 6.10 | 6.67 | 1.37 |
> | EC-zh | 6.10 | - | 1.70 | 5.31 | 0.93 |
>
> **Findings**:
>
> - At comparable model sizes, dedicated RP models generally outperform general LLMs, particularly in emotional intelligence during dialogue (CA) and human-like logical reflection (RR). In Chinese, Qwen3-8B shows impressively strong RP performance, potentially because the Qwen series has explicitly optimized role-playing as a core capability since Qwen2.5 (mentioned in its technical report).
>
> - Across all leaderboard, dedicated RP models remain substantially weaker than the top performers (o3 and deepseek-r1), and the gap generally stems from: 1) very large differences in parameter scale and fundamental intelligence; 2) role-playing has become an explicit optimization target in modern LLM training; 3) stronger generalization ability in current SOTA general LLMs.
>
> - Interestingly, dedicated RP models exhibit lower RP hallucination rates than general LLMs of similar scale. For example, Humanish-Roleplay-Llama-3.1-8B on the English Synthesized Character subset even outperforms the top performers. We believe this is primarily due to overfitting on the RP task, and there is no explicit risk of cross-domain goal conflict
>
> Finally, **we would clarify why the initial submission did not include dedicated RP models**: our preliminary experiments (e.g., on CoSER-7B and CoSER-70B) showed that compared with current powerful general LLMs, dedicated RP models often lack generalization and perform well only under specific RP evaluation setups. In contrast, our goal with FURINA is to provide a general, challenging, and capability-oriented RP benchmark that evaluates modern frontier models.

---

> ### Author Response · Authors · 2025-11-26
> **Response to Reviewer MCg3 (2/2)**
>
> > Analysis on Reasoning Utility
>
> We thank the reviewer for pointing out this concern, and we think the selected main results in Table 4 can provide a detailed qualitative and quantitative analysis regarding the role of reasoning.
>
> | Model | Context Reliance (CR) | Factual Recall (FR) | Reflective Reasoning (RR) | Conversational Ability (CA) | Preference Alignment (PA) | Weighted Average Score |
> |---|---|---|---|---|---|---|
> | **English** |
> | Qwen3-32B | 15.98 | 13.44 | 9.08 | 15.29 | 20.21 | 14.80 |
> | Qwen3-32B-T | 28.49 | 27.47 | 13.90 | 27.72 | 38.78 | 27.27 |
> | Qwen3-235B | 19.61 | 18.99 | 19.88 | 23.16 | 32.23 | 22.77 |
> | Qwen3-235B-T | 27.37 | 35.45 | 21.88 | 28.08 | 45.50 | 31.66 |
> | Claude-4-Sonnet | 32.62 | 27.98 | 48.30 | 30.41 | 41.74 | 36.21 |
> | Claude-4-Sonnet-T | 30.80 | 29.72 | 53.45 | 27.71 | 40.90 | 36.52 |
>
> **Findings**:  Reasoning generally improves model performance on RP tasks, but the mechanisms of improvement differ across models. For **Qwen3**, reasoning serves as a robust enhancement, improving performance across multiple dimensions including contextual consistency (CR), fundamental world knowledge (FR), human-like reflection (RR), conversational intelligence (CA), and dialogue tone (PA). For **Claude-4-Sonnet**, however, the improvements from reasoning are predominantly concentrated in the FR and RR dimensions, while performance in the remaining dimensions even shows signs of decline. This discrepancy is likely attributable to the distinct reasoning training approaches, where more generalized reasoning algorithms or exposure to RP-related data during reasoning training tend to yield more reliable improvements in RP tasks.
>
> We sincerely hope our responses could address your concerns, but feel free to let us know if you have any other questions!

---

> > ### Comment · Reviewer_MCg3 · 2025-11-27
> >
> > Thank you for the authors' response, which has addressed some of my concerns. In light of this, I have raised my score to 4. However, the overall quality of the paper still falls below my expectations.

---

### Official Review · Reviewer_629i · 2025-10-30

**Soundness:** 2
**Presentation:** 2
**Contribution:** 1
**Rating:** 2
**Confidence:** 4

**Summary:**

The paper proposes the FURINA-Builder, a pipeline to create role-playing benchmarks, and the FURINA-Bench, a specific benchmark created with this system.

The actors:
- Evaluated model: $M_{source}$.
- Baseline model: $M_{base}$, GPT-4.1 in FURINA-Bench.
- Judge model: $M_{judge}$, GPT-4.1 in FURINA-Bench.
- Scene character model: $M_{scene}$, Qwen3-235B-A22B in FURINA-Bench.
- Director model: $M_{director}$, Qwen3-235B-A22B in FURINA-Bench.
- A model that checks hallucination-related keywords in  $M_{judge}$ CoT: $M_{checker}$

The pipeline is the following:
1. The set of characters with public/private attributes is defined. Characters might be either manually or automatically generated.
2. Scenarios are extracted from a set of books.
3. A test character is inserted in a scenario, where it communicates with the original characters from this scene, played by $M_{scene}$.
4. $M_{director}$ decides which character speaks in the current conversation turn. It might be $M_{scene}$ or $M_{source}$ + $M_{base}$.
5. If it is a test character, the utterance is generated both by the evaluated model $M_{source}$ and by the baseline model $M_{base}$.
6. Both utterances are scored by $M_{judge}$. First, it selects the most relevant evaluation dimension for the current turn. Second, it outputs a number from the 5-point Likert scale, where 1 indicates a strong preference for $M_{source}$ and 5 indicates a strong preference for $M_{base}$, judging by the selected evaluation dimension. If the score is 4 or 5, the $M_{base}$ response is stored in the conversation history.
7. Finally, all the scores are aggregated and normalized, and this is how the final leaderboard appears.

The FURINA-Bench specifics:
- 2 languages: English and Chinese
- At least 10 scores for each character and each dimension
- Specific models (see above)
- 5 specific evaluation dimensions
- 20 test characters, ~1500 conversations, ~7000 test utterances

Human validation:
- Accuracy of $M_{judge}$ dimension selection (with 5 specific evaluation dimensions), 1000 samples.
- Pearson correlations of $M_{judge}$ scores with human scores in 5 specific evaluation dimensions, 400 samples.

Results:
- Bigger models are better.
- Reasoning models are also better.
- The model receives better scores when provided with established characters instead of synthesized.
- "Reasoning improves RP performance but amplifies hallucinations."

**Strengths:**

Overall, the paper is well organized. Figures are clear, and the appendices are extremely comprehensive (prompts, examples, annotation UI). It's nice that it has evaluations not only for English.

The pipeline makes sense, and the actors are clearly defined. Overall, the design of the self-play framework is reasonable.

**Weaknesses:**

Major points:

1. The main weakness: it's not clear what problem this pipeline/benchmark actually solves. Usually, in all role-playing applications, models interact with **users**. The primary complexity in building a role-playing benchmark lies in user simulation. This paper just removes users from the picture. The remaining self-playing framework evaluates whether models are good at interacting with each other. But why should anyone care about that?

2. Another significant problem is in the positioning as a "first RP benchmark builder". It claims that users can customize a set of characters, scenarios, and evaluation criteria. However, isn't it the case for almost any other open source benchmark?

3. Authors claim that they "conduct human evaluations, which confirm the builder’s effectiveness and benchmark’s soundness". But all human evaluations use the specific evaluation dimensions from a specific benchmark. So they just can't validate "builder’s effectiveness". Overall, the "builder" was used to build only a single benchmark.

4. The evaluation schema is a standard pairwise schema. For instance, it was used in the [RPBench-Auto](https://www.boson.ai/blog/rpbench-blog). Overall, I think that the originality of the approach is very limited.

5. The paper is too verbose on non-important things and not verbose enough on important things. For instance, what is the inter-annotator agreement? Without it, the reported accuracies/correlations are hard to interpret.

Minor points:

1. The section about the "emerging Pareto frontier" is very weak. It "emerges" only because of the single outlier, GPT-4o.

2. The whole "hallucination" story is not validated in any way. The accuracy of the hallucination checker is not reported.

3. The evaluation model is always GPT-4.1, and there are no ablations or robustness checks about that.

4. There are potential leaks in the evals: Qwen3-235B-A22B is both source/director/scene, and it is reported in the leaderboard table. This could advantage models in contexts in their own style and violates the strict independence of eval data.

5. The procedure of inserting the base model utterance instead of the target model utterance if the judge's score is 4 or 5 is questionable. Measuring cascading errors is important, and it is not clear why we would want the history to always be perfect.

**Questions:**

See the questions in the "Weaknesses" section.

Additional questions:

1. What is "CustomRPBench"? It appears in many figures. Sounds like a deprecated name.

2. How were the models selected for evaluation? For instance, why are there no Gemma models?

**Details Of Ethics Concerns:**

The paper uses copyrighted materials (books). Human annotators were employed with undisclosed compensation.

---

> ### Author Response · Authors · 2025-11-26
> **Response to Reviewer 629i (1/3): Main Points**
>
> We greatly appreciate the reviewer’s detailed feedback and suggestions. We would like to first address the reviewer’s main high-level concerns (Major points), and then respond to each minor point and additional question individually.
>
> > The main weakness: it's not clear what problem this pipeline/benchmark actually solves. This paper just removes users from the picture. The remaining self-playing framework evaluates whether models are good at interacting with each other. But why should anyone care about that?
>
> The concern that our benchmark "removes users" stems from a key **misunderstanding**. Rather than narrowing the scope, we establish a generalized evaluation environment where every participant is a character with a distinct persona, causing the distinction between a "user" and a "character" to vanish. **Consequently, whether an LLM interacts with a human user or an LLM-based character does not fundamentally alter the evaluation of its core abilities, such as persona consistency and emotional intelligence.** Furthermore, our approach is also vital and necessary for benchmarking LLMs in many important scenarios, like group chat scenarios, game NPCs, and social simulations.
>
> > Another significant problem is in the positioning as a "first RP benchmark builder". It claims that users can customize a set of characters, scenarios, and evaluation criteria. However, isn't it the case for almost any other open source benchmark?
>
> Previous RP benchmarks provide **fixed datasets** where evaluated LLMs must role-play **predetermined characters** with **static persona prompts** under **pre-defined evaluation dimensions**. However, in real applications, performance on fixed benchmarks does not guarantee success in your specific use case — this transferability gap becomes particularly pronounced when comparing similarly-performing models. FURINA-Builder addresses this by providing **an automatic pipeline** that allows users to **fully customize role definitions**, **scenario pools**, **dialogue structures**, and evaluation dimensions. This enables practitioners to construct benchmarks closely aligned with their actual deployment scenarios, yielding more reliable performance predictions than generic benchmarks.
>
> > Authors claim that they "conduct human evaluations, which confirm the builder’s effectiveness and benchmark’s soundness". But all human evaluations use the specific evaluation dimensions from a specific benchmark. So they just can't validate "builder’s effectiveness". Overall, the "builder" was used to build only a single benchmark.
>
> We respectfully disagree with the characterization that building a single benchmark cannot validate our builder's effectiveness. Our validation strategy is methodologically sound: we demonstrate the builder's capability by constructing FURINA-Bench with deliberately general character personas and evaluation dimensions, then validate through human annotation that the resulting benchmark produces meaningful, reliable assessments. **More importantly, the effectiveness of a builder should be assessed by its ability to handle diverse roles and evaluation dimensions, rather than simply counting the number of benchmarks produced**. Our single benchmark encompasses multiple character types and varied evaluation criteria, which more convincingly demonstrates the builder's versatility and robustness than creating multiple benchmarks, each with narrow configurations.
>
> > The evaluation schema is a standard pairwise schema. For instance, it was used in the RPBench-Auto. Overall, I think that the originality of the approach is very limited. The paper is too verbose on non-important things and not verbose enough on important things.
>
> Our main contribution is the introduction of the FURINA-Builder concept and pipeline, as well as a systematic analysis of the resulting FURINA-Bench, which is exactly where the paper focuses on the majority of its exposition. The other reviewers also agreed on our main contributions. We do not claim originality for the standard pairwise evaluation schema, which is a simple yet effective practice already established in current RP evaluation.
>
> We hope that through these explanations, the reviewer can recognize the value and contribution of our work in the RP field.

---

> ### Author Response · Authors · 2025-11-26
> **Response to Reviewer 629i (2/3): Minor Points**
>
> > Concerns about "Pareto frontier constraining"
>
> The goal of proposing this concept "RP Pareto frontier" is primarily to help readers understand the challenges in current RP systems more easily: RP performance and RP reliability cannot be optimized simultaneously. While the reviewer suggests that the frontier appears only because of GPT-4o, our results show otherwise. **Claude4-Thinking** in the right part of Figure 4 also lies near the boundary and helps define the same trade-off.
>
> Such sparsity is mainly due to the limited number of evaluated models. After incorporating a few additional dedicated RP models following Reviewer MCg3’s suggestion, we found another model - **Peach-2.0-9B-8k-Roleplay** - also aligns with the proposed frontier with the coordinate **(x, y) = (16.93, 58.82)** in the left part of Figure 4. This also indicates the presence of models with high RP reliability but relatively low RP performance. It would be clearer to see the updated Figure 4.
>
> > Accuracy of the hallucination checker is not reported
>
> We thank the reviewer for highlighting this potential gap. Since the hallucination detection task is very simple - **keyword detection on Judgement CoT using a 32B instruction model** - we conducted only a preliminary feasibility study to design a reliable prompt (Appendix T), rather than a full-scale manual validation. We have already started a human annotation procedure to quantify the accuracy of the checker. However, due to administrative constraints, we cannot guarantee completion within the rebuttal period. We promise that we will include the annotated results in the Appendix of the final version.
>
> > Concerns about Judge Model
>
> In **Table 3**, we already reported the Pearson correlations and p-values between model scores and human annotations, across different models (**GPT-4.1**, **DeepSeek-R1** and **Deepseek-V3**) and on **all evaluation dimensions**.  Our results show that GPT4.1 is a relatively robust judge model in our task.
>
> > Potential leaks in the evals for Qwen3-235B-A22B
>
> We thank the reviewer for pointing out this potential issue. When we used the FURINA-Builder to create FURINA-Bench, we selected a wide range of LLMs as test characters. According to **Figure 7** and **Figure 8**, only **4.3%** and **4.2%** of the source models in our dataset come from Qwen3-235B-A22B, representing only **a quite small portion**. Moreover, in our evaluation we included many models, with Qwen3-235B-A22B being just one of them. It is neither a SOTA model nor central to our main analysis, so we believe the potential impact is **minimal**.
>
> > Concerns about dialogue history quality
>
> **In general, evaluating a model’s RP performance will become very difficult if the dialogue history is contaminated with errors or incoherent turns**. For example, in such cases, the most appropriate response might trivially become “What is everyone talking about?”. Ensuring that the dialogue history remains as clean as possible helps minimize confounding factors and allows us to focus on assessing the model’s RP capabilities directly.

---

> ### Author Response · Authors · 2025-11-26
> **Response to Reviewer 629i (3/3): Other Questions**
>
> > What is "CustomRPBench"? It appears in many figures. Sounds like a deprecated name.
>
> We appreciate the reviewer pointing this out carefully. CustomRPBench is indeed a deprecated name that was not fully updated throughout the figures. We have already corrected all remaining occurrences in the new revision.
>
> > How were the models selected for evaluation? For instance, why are there no Gemma models?
>
> We aimed to include strong models that are widely recognized within the RP community. **However, it is not feasible to evaluate all available models due to space limitations and the substantial cost**. Regarding **Gemma**, we indeed evaluated it. However, we observed that the Gemma API occasionally blocks certain keywords, producing unnatural or disrupted RP outputs. For this reason, we moved the results to **Appendix N** and noted this in the **caption of Table 4**.
>
> > Ethics Concerns: The paper uses copyrighted materials (books). Human annotators were employed with undisclosed compensation
>
> For copyright considerations, we already mentioned in the **ETHICS STATEMENT** section: "In addition, we strictly adhere to copyright policies and do not distribute any raw novel content, and we require that any users of our work obtain proper permissions when creating derivative resources". Here we further **highlight** that all data in our dataset has been adjusted and transformed by LLMs, and is intended solely for research purposes. As the FURINA-Bench dataset is derived from literary material, it may involve ethical considerations, and the content presented does not reflect the views of the original authors. **Pretty similar processing and statement can be found in ICML 2025 CoSER: Coordinating LLM-Based Persona Simulation of Established Roles**.
>
> Our human annotators were compensated at levels exceeding typical market rates, in accordance with fair-labor and ethical research standards.
>
> We have already updated all of them in the new **ETHICS STATEMENT** section.

---

### Official Review · Reviewer_wCx3 · 2025-10-31

**Soundness:** 3
**Presentation:** 4
**Contribution:** 3
**Rating:** 6
**Confidence:** 3

**Summary:**

This paper addresses the limitations of existing role-playing (RP) benchmarks, which are often static, narrow in scope, and quickly become obsolete. The authors introduce FURINA-Builder, a novel multi-agent collaboration pipeline designed to automatically construct fully customizable and scalable RP benchmarks. This pipeline leverages a well-constructed character-scene pool for simulation, using a Director Model and a Judge Model for dynamic selection during dialogue generation. This process yields diverse, high-quality dialogue data annotated with fine-grained evaluation dimensions.
Using this builder, the authors develop FURINA-Bench, a new comprehensive RP benchmark that, for the first time, integrates both "established" and "synthesized" characters within group chat scenarios. Extensive evaluations of cutting-edge LLMs on this benchmark reveal several key insights: 1) o3 and DeepSeek-R1 achieve the best performance on English and Chinese RP tasks, respectively. 2) The reasoning capabilities of models, while improving RP performance, also significantly increase RP hallucinations. 3) The paper uncovers a Pareto frontier between RP performance and reliability (low hallucination rate), which has significant implications for future RP model development.

**Strengths:**

1. Novelty of Methodology: The primary contribution is FURINA-Builder itself—a shift from creating a static benchmark to providing a dynamic benchmark builder. This is a significant conceptual advance in the RP evaluation space, effectively addressing the fundamental problem that existing benchmarks (like CoSER, CharacterBench) struggle to adapt to new characters, scenarios, and rapidly evolving LLM capabilities. This "benchmark for benchmarks" design is highly forward-looking.
2. Sophisticated Pipeline Design: The multi-agent collaboration pipeline of FURINA-Builder is well-designed. The "Selection" phase (Figure 1) is particularly clever: the Judge Model first determines the most appropriate evaluation dimension (Context Reliance, Factual Recall, etc.) for the current turn before comparing the source (Msource) and base (Mbase) models. This mechanism not only ensures high-quality dialogue trajectories (by filtering out inferior responses) but also automatically labels each data point in the benchmark with its most relevant, fine-grained evaluation criterion, achieving two goals at once.
3. Comprehensive Benchmark and Insights: The resulting FURINA-Bench is itself a high-quality resource. It is the first to unify the evaluation of "established" and "synthesized" characters in a multi-character group chat setting, which is crucial for simulating realistic virtual worlds (e.g., game NPCs). More importantly, the experimental analysis based on this benchmark yields valuable and novel conclusions. The discovery of the Pareto frontier between RP performance and reliability (Figure 4) and the non-intuitive trade-off that "reasoning capabilities exacerbate hallucinations" (Figure 2) are both critical findings.

**Weaknesses:**

1. Heavy Reliance on LLM-as-a-Judge: The entire FURINA-Builder pipeline (for building the benchmark) and the FURINA-Bench evaluation protocol (for testing models) are heavily dependent on an LLM (specifically, GPT-4.1) as the judge. Although the authors provide human validation in Section 4.4 (Tables 2 & 3), the results (e.g., Pearson correlations around 0.6 in Table 3) indicate that the LLM judge is only moderately, not perfectly, aligned with human judgment. This implies that the benchmark's data quality and the subsequent evaluation results may systematically inherit the biases of GPT-4.1. For instance, it remains questionable whether GPT-4.1 can truly and accurately assess more subjective dimensions like "Reflective Reasoning" (RR) or "Preference Alignment" (PA).
2. Concerns on Cost and Scalability: The authors claim FURINA-Builder can construct benchmarks "at any scale," but the multi-agent process appears computationally expensive. Generating a single test utterance requires multiple calls to powerful LLMs (Director, Mscene, Msource, Mbase, Mjudge). The paper provides no estimation of the computational resources (e.g., API costs or token consumption) required to build FURINA-Bench. This potentially high cost could significantly hinder the community from adopting FURINA-Builder to create their own "customizable" benchmarks, thereby weakening the practical utility of the core "builder" contribution.
3. Oversimplified Definition of RP Hallucination: In Section 5.4, the paper divides RP hallucinations into "EC hallucination" (a factuality hallucination, measured by the FR dimension) and "SC hallucination" (a faithfulness hallucination, measured by the CR dimension). This demarcation seems overly simplistic and arbitrary. An "established character" (EC) can certainly produce hallucinations unfaithful to the current context (CR), just as a "synthesized character" (SC) can fabricate real-world facts (FR). Forcibly binding character types to specific hallucination types and measuring them via only two evaluation dimensions likely fails to capture the full spectrum of RP hallucinations (e.g., the more common OOC - Out of Character).

**Questions:**

See weaknesses.

---

> ### Author Response · Authors · 2025-11-26
> **Response to Reviewer wCx3**
>
> We appreciate the reviewer’s close reading of the paper and that the reviewer found our work innovative and meaningful. We respond to the reviewer’s questions below.
>
> > Reliance on LLM-as-a-Judge
>
> While we acknowledge that LLM-based evaluation sometimes cannot perfectly align with human judgment, a Pearson correlation **above 0.6** is typically considered a strong correlation in the current research area (e.g., McHugh, Interrater Reliability: The Kappa Statistic, 2012). This suggests that in our paper, GPT-4.1 achieves reasonably strong consistency with human annotators.
>
> > Concerns on Cost and Scalability
>
> We thank the reviewer for raising this practical concern. **The goal of FURINA-Builder is precisely to enable customizable benchmark construction with flexible model choices.** Although we used large SOTA models to produce a general-purpose challenging benchmark for public release, users are fully free to replace any module (Director, Mscene, Msource, Mbase, Mjudge) with any models (like smaller open-source models) according to their resource boundary. At the same time, we open-source a well-constructed scene-character pool to provide convenience for the community.
>
> > Definition of RP Hallucination
>
> We appreciate the reviewer’s detailed attention to the nuances of RP hallucination! Our distinction between **EC (Established Character)** and **SC (Synthesized Character)** hallucinations is definition-driven and grounded in model behavior, not arbitrary. **For ECs**, persona information is already internalized in the model parameters, so persona violations will naturally manifest in the Factual Recall (FR) dimension. Since character persona does not need to be repeatedly specified in the context window, inconsistencies in CR dimension for ECs usually reflect logical or dialogue-flow issues rather than RP hallucinations. **For SCs**, the model has never seen these character personas before; thus, RP hallucinations can no longer arise from FR, and may occur only in the Context Reliance (CR) dimension.
>
> Overall, the logic here is that we **first** formally define these two types of RP hallucinations, and **then** evaluate each only within its corresponding dimension. Detecting them in unrelated dimensions could capture non-RP hallucinations and confound the evaluation results.

---

### Author Response · Authors · 2025-12-01
**General Summary**

Dear Reviewers and AC,

We sincerely appreciate the reviewers' time and effort in reading our paper and giving their thoughtful feedback, as well as AC's time for meta-review. We are pleased that the reviewers think our proposed FURINA-Builder is innovative  [```wCx3```, ```MCg3```] and reasonable [```629i```], our constructed FURINA-Bench is comprehensive and high-quality [```wCx3```, ```MCg3```], and the experimental analysis on the benchmark is insightful [```wCx3```].

For the main concern about what problem this pipeline/benchmark actually solves raised by reviewer ```629i```, we believe this **key misunderstanding** arises because the reviewer focuses primarily on traditional human–LLM interaction settings and is not very familiar with modern role-playing scenarios. We already replied to these high-level concerns one by one with concise and powerful responses. In contrast, both Reviewer ```wCx3``` and Reviewer ```MCg3``` recognized the value of our work in the RP area and reached a clear consensus on the specific contributions we provide.

We have carefully addressed all reviewer comments point by point and incorporated the corresponding revisions into the updated manuscript. Below, we provide a summary of the major changes and newly added experiments in the latest paper version.

1. Expanded related work [```MCg3```]. We added a new prominent dataset RoleBench in **Table 1** and cited Neeko training method in **Section 1** for better user understanding.

2. Evaluation of dedicated RP LLMs [```MCg3```, ```629i```].  We evaluated additional dedicated RP models and report results in **Table 4**, **Table 5**, and **Figure 4**, with analysis comparing their characteristics to general LLMs in **Section 5.2 and 5.5**.

3. Deeper analysis of reasoning utility [```MCg3```]. We expanded experiments and discussion that investigated how reasoning capabilities contribute to RP performance in **Section 5.3**.

4. Clarified motivation and pipeline design choices [```629i```]. We made our motivations and pipeline design decisions more explicit in **Section 3.4 and 4.4**.

5. Clear ethics statement [```629i```]. We provide a more comprehensive and clear ethics statement in **Section 7**.

All suggested modifications have been incorporated into the latest manuscript and are highlighted in **blue**. We are grateful for the opportunity to improve our work and also thank you, AC, for taking the time to fully meta-reviewing our paper during this special time.

Best regards,

The Authors

---

### Note · Authors · 2025-12-15

I have read and agree with the venue's withdrawal policy on behalf of myself and my co-authors.